# Boosting Multiagent Reinforcement Learning at High Replay Ratios with Ensemble Reset

## Abstract

Reinforcement learning with a high replay ratio, where the agent's network parameters are updated multiple times per environment interaction, is an emerging way to improve sample efficiency. However, this paradigm remains underexplored in multiagent reinforcement learning (MARL). In this paper, we investigate how to efficiently train MARL at high replay ratios to accelerate learning. Surprisingly, we found that simply increasing the replay ratio leads to severe dormant neurons in the centralized global Q-value network, where neurons become inactive thereby undermining network expressivity and hindering the learning of MARL. To tackle this challenge, we propose Ensemble Reset (EnSet) to boost MARL at high replay ratios from two aspects. First and for the first time, EnSet utilizes an ensemble of global Q-value networks with parameter reset to reduce dormant neurons when updated at a high frequency. Second, EnSet diversifies the replay experience using a multiagent translation invariance prior of the global Q-function to prevent overfitting. Extensive experiments in SMAC, MPE, and SMACv2 show that EnSet substantially speeds up various MARL algorithms at high replay ratios.

## 1 Introduction

In recent years, multiagent reinforcement learning (MARL) has achieved significant advances in diverse domains that rely on collective intelligence, where multiple agents interact to solve complex sequential decision-making problems. Notable successes include real-time strategy games (Berner et al., 2019; Vinyals et al., 2019), distributed energy management (Yang et al., 2019; Novati et al., 2021), urban traffic signal control (Wu et al., 2020; Ma et al., 2024), etc. When applying MARL to real-world tasks, a major drawback of existing approaches is their low sample efficiency, which requires a huge amount of environmental interactions to learn satisfactory policies.

Although sample-efficient MARL algorithms are critical when deployed in the real world, where interaction with the environment is often limited, improving sample efficiency in the field of MARL remains a longstanding challenge without achieving a substantive breakthrough. Most existing works leverage the multiagent permutation prior to improve sample efficiency. For example, Ye et al. (2022) generate additional data by applying a permutation transform to homogeneous agents, exploiting the permutation invariance. In addition, Yu et al. (2023) utilize the global system symmetry to augment data, where rotating the global state induces a permutation of the optimal joint policy. Later, Hao et al. (2023) incorporate both permutation invariance and permutation equivariance inductive biases into agent network design to boost MARL. Orthogonal to the above works, another line of research starts to increase the replay ratio in MARL to improve sample efficiency. The replay ratio is defined as the number of updates of agent network parameters per environment interaction, and increasing it is an appealing strategy for improving sample efficiency by performing more updates within a fixed interaction budget (Chen et al., 2021; D'Oro et al., 2023). Recently, Yang et al. (2024) and Xu et al. (2024) found that increasing the replay ratio is efficient to accelerate the learning of MARL. Notably, Yang et al. (2024) periodically reset networks to mitigate the plasticity loss (Nikishin et al., 2022; Lyle et al., 2023) when training MARL at high replay ratios.

In this paper, we delve into boosting MARL with a high replay ratio to further push the boundary of sample efficiency from a novel angle centered on the global Q-network. Accordingly, we develop a novel method named Ensemble Reset (EnSet) to improve MARL sample efficiency by two innovations: (1) global Q-network ensemble reset and (2) global Q-value translation invariance. First and

for the first time, we reveal that the centralized global Q-network of MARL suffers from a severe dormant neuron issue at high replay ratios, where a large portion of network neurons become inactive to undermine network expressivity. Fortunately, this issue is effectively mitigated by the network ensemble. Based on this finding, EnSet employs an ensemble of global Q-networks combined with periodic parameter reset to reduce dormant neurons for stabilizing training. Second, EnSet diversifies the replay experience to prevent overfitting in the high-replay-ratio setting using global Q-value translation invariance, where a system-level coordinate translation of units results in the same global Q-values. Extensive experiments on a total of 14 tasks in SMAC, MPE, and SMACv2 environments demonstrate that EnSet significantly accelerates multiple MARL algorithms at high replay ratios with far fewer environment interactions, while beating various network reset baselines.

## 2 BACKGROUND

### 2.1 MARKOV GAMES

We use Markov games as the basic setting, which are an extended multiagent version of Markov Decision Processes (Littman, 1994). Markov games are described by a state transition function, $T : S \times A_1 \times ... \times A_N \rightarrow P(S)$, which defines the probability distribution over all possible next states given the current global state $s \in S$ and each agent $i$'s action $a_i \in A_i$. The reward is given based on the global state and actions of all agents $R_i : S \times A_1 \times ... \times A_N \rightarrow \mathbb{R}$. Meanwhile, Markov games can be partially observable, where each agent $i$ perceives the environment through a local observation function $o_i = \mathcal{O}(s, i) \in O_i$, where $O_i$ is agent $i$'s observation space. Consequently, each agent learns a policy $\pi_i : O_i \rightarrow P(A_i)$ that maps each agent's observation to a probability distribution over its action set, to maximize this agent's expected discounted cumulative returns, $J_i(\pi_i) = \mathbb{E}_{a_1 \sim \pi_1, ..., a_N \sim \pi_N, s \sim T}[\sum_{t=0}^{\infty} \gamma^t R_i(s_t, a_{1,t}, ..., a_{N,t})]$, where $\gamma \in [0, 1)$ is the discounted factor. If all agents share an identical reward function, i.e., $R_1 = ... = R_N$, Markov games are fully cooperative (Matignon et al., 2012): a best-interest action of one agent is best-interest for others.

Many representative off-policy MARL algorithms, such as QMIX (Rashid et al., 2018) and MAD-DPG (Lowe et al., 2017), adopt the centralized training with decentralized execution (CTDE) paradigm to learn agent policies. CTDE allows agents to access global information during training for better coordination and to act independently based on local observations only when deployed for execution. In CTDE, there is usually a centralized global Q-value network to evaluate the joint policy given the multiagent system state and provide gradients to update each agent's decentralized policy, ensuring agents learn cooperative behaviors. In this work, we follow the CTDE paradigm.

### 2.2 REPLAY RATIO

Replay ratio refers to the number of updates of an agent's parameters for each environment interaction (Chen et al., 2021; D'Oro et al., 2023). As each interaction with the environment comes at a cost, it is desirable to perform more updates with existing experiences before interacting with the environment again. Therefore, increasing the replay ratio is an appealing strategy for improving the sample efficiency of deep reinforcement learning (Chen et al., 2021; D'Oro et al., 2023; Schwarzer et al., 2023; Nauman et al., 2025; Lee et al., 2025), which has received much attention nowadays. Among them, Kim et al. (2023) utilize a sequential reset ensemble and adaptive agents composition to mitigate performance collapses caused by resetting at high replay ratios in the domain of safe RL.

To quantify sample efficiency and clarify the critical role of the replay ratio, we adopt the definition from Ye et al. (2022), which formalizes the expected number of times each experience in the replay buffer is sampled for agent training as

$$\mathbb{E}[N_{sampled}] = \frac{N_{RR} \cdot N_B}{V \cdot T_U}, \tag{1}$$

where $N_{RR}$ is the number of replay ratios for updating, which is performed $N_{RR}$ times when interacting with the environment once. $N_B$ is the batch size that there are $N_B$ data experiences sampled into a batch for updating. The data acquisition speed $V$ is the number of transition data experiences that are collected at each time step of environment interaction. $T_U$ is the update interval where the updating is conducted every $T_U$ time steps. Typically, works focusing on training reinforcement learning agents at a high replay ratio (Chen et al., 2021) aim to increase $N_{RR}$ while keeping $N_B$,

$V$, and $T_U$ unchanged. They seek to achieve higher sample efficiency through a higher $\mathbb{E}[N_{sampled}]$ with different approaches, finally improving performance with the same number of environment interactions, i.e., the same number of collected data experiences. This behavior is also termed replay ratio scaling (D'Oro et al., 2023), which is much less studied in the MARL domain than single-agent reinforcement learning. Until recently, Yang et al. (2024) and Xu et al. (2024) found that properly increasing the replay ratio improves the sample efficiency of MARL algorithms.

### 2.3 THE DORMANT NEURON PHENOMENON

Sokar et al. (2023) identify the dormant neuron phenomenon in deep reinforcement learning, where an agent's network suffers from an increasing number of inactive neurons during the training process, thereby affecting network expressivity. The definition of the dormant neuron is given below.

**Definition 1** ($\alpha$-Dormant Neuron). Given an input distribution $D$, let $\rho_j^l(x)$ denote the activation of neuron $j$ in layer $l$ under input $x \in D$ and $N_l$ be the number of neurons in layer $l$. The normalized activation score $d_j^l$ of a neuron $j$ in layer $l$ is defined as follows:

$$d_j^l = \frac{\mathbb{E}_{x \in D}|\rho_j^l(x)|}{\frac{1}{N_l}\sum_{k=1}^{N_l}\mathbb{E}_{x \in D}|\rho_k^l(x)|}. \tag{2}$$

Then neuron $j$ in layer $l$ is defined as $\alpha$-dormant if its activation score $d_j^l \leq \alpha$. In this paper, following Qin et al. (2024) who study the dormant neurons in MARL algorithms, we set $\alpha$ at 0.1. The dormant neuron rate for a neural network is defined as the proportion of the $\alpha$-dormant neurons.

A few works have started to study the dormant neuron phenomenon in the domain of MARL. For example, Xu et al. (2024) use the dormant neuron rate as an indicator of plasticity when training MARL at high replay ratios. They found that the dormant neuron rate in the agent's policy network remains at a low level during training due to the agent's RNN structure. However, they do not consider the dormant neuron in the global Q-network, which is the core module in CTDE. More recently, Qin et al. (2024) found that, in multiagent value factorization algorithms, dormant neurons in the global Q-network increase as training proceeds with a standard replay ratio of 1. Accordingly, they propose ReBorn, which transfers the weights from overactive neurons to dormant neurons to help learning. However, they neglect to further study under a high replay ratio, leaving the effective scaling of the replay ratio for better sample efficiency unexplored. Another recent work is MARR (Yang et al., 2024), which investigates a closely related concept, plasticity loss (Lyle et al., 2023), in MARL. MARR introduces a multiagent Shrink & Perturb strategy to periodically reset network parameters to maintain the network plasticity. The multiagent Shrink & Perturb in MARR is defined as

$$\theta_t^i \leftarrow \alpha_r \theta_t^i + (1 - \alpha_r)\theta_0^i, \text{ for } i = 1, 2, ..., N, \tag{3}$$

and

$$\phi_t \leftarrow \alpha_r \phi_t + (1 - \alpha_r)\phi_0, \tag{4}$$

where $\theta_0^i$ is agent $i$'s initial policy or individual Q-value network parameters. $\phi_0$ is the initial global Q-network parameters. $\theta_t^i$ and $\phi_t$ are the current agent and global Q-value network parameters, respectively. The interpolation factor $\alpha_r$ decides how much the current network parameters are kept. However, none of these works explicitly investigates the connection between dormant neurons in the global Q-network and high replay ratios, which is the key bottleneck to MARL sample efficiency.

## 3 BOOSTING MARL AT REPLAY RATIO VIA ENSEMBLE RESET

In this section, we introduce the proposed EnSet, which is illustrated in Figure 1. First, we found that the dormant neuron phenomenon in the global Q-value network is exacerbated when improving the training replay ratio in Section 3.1. Then, in Section 3.2, we show that the network ensemble, which employs multiple global Q-networks, surprisingly mitigates the phenomenon of dormant neurons and helps stabilize learning, especially when combined with the periodic network reset. Moreover, we introduce the global Q-value translation invariance to augment the global state in Section 3.3.

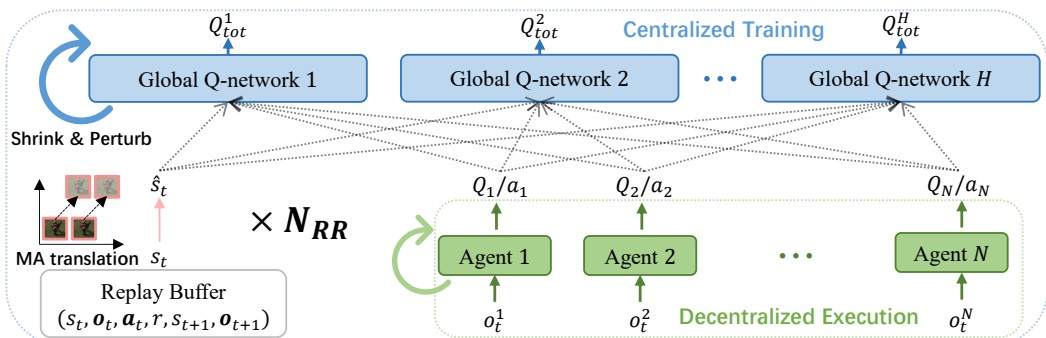

Figure 1: The illustration of the EnSet framework, which is trained under a high replay ratio of $N_{RR}$. The global Q-network ensemble with network reset tackles dormant neurons under high $N_{RR}$. The global state translation invariance is introduced to diversify the state $s$ of the global Q-value function.

## 3.1 DORMANT NEURONS IN THE GLOBAL Q-NETWORK AT HIGH REPLAY RATIOS

First, we show how the dormant neuron in the global Q-network correlates with the replay ratios, which have never been explicitly studied before to the best of our knowledge. We use QMIX (Rashid et al., 2018), one of the most popular MARL algorithms, as the tested base algorithm. Following Qin et al. (2024), we measure dormant neurons in the hypernetworks of QMIX's global Q-network. In the standard setting, the replay ratio is 1, which means that the network parameters are updated once after one episode. We increase the replay ratio to 5 and 10. The dormant ratio rates in the centralized Q-value network of QMIX under different replay ratios are given in Figure 2(a) and 2(b). As we can see, the dormant neuron phenomenon becomes severe enough to hinder the learning of QMIX in high-replay-ratio settings. This finding motivates us to reduce the dormant neurons to increase the replay ratio for boosting the sample efficiency of MARL algorithms. More cases as well as the dormant neuron rates in the agent network maintaining a low level are shown in Appendix A.

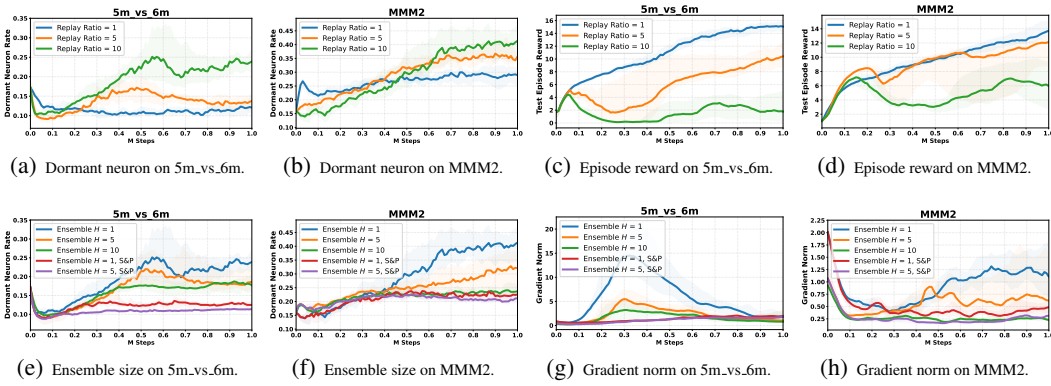

Figure 2: (a-b) show that the dormant neuron rate in the global Q-network is exacerbated at high replay ratios. (c-d) show that, when the replay ratio is 10, the learning collapses with many dormant neurons being inactive. (e-f) show that dormant neurons in global Q-networks are mitigated by the network ensemble as well as multiagent Shrink & Perturb (denoted as S&P) when the replay ratio is 10. (g-h) show that the ensemble smooths the updating gradient for network parameters.

## 3.2 GLOBAL Q-NETWORK ENSEMBLE WITH NETWORK RESET

Next, we show that the network ensemble is an efficient way to tackle the severe dormant neuron phenomenon when training at high replay ratios. For the first time, we propose Ensemble Reset to utilize an ensemble of global Q-networks with network parameter reset to stabilize MARL training at high updating frequency. As shown in Figure 1, we employ $H$ global Q-networks in the central-

ized training stage. In contrast, we do not enable ensemble on agent policy or individual Q-value networks, as previous studies (Xu et al., 2024; Qin et al., 2024) show that no severe dormant neuron phenomenon exists in agent networks. Additionally, it ensures that, in the decentralized execution stage, no extra storage or computation resources are needed to deploy the learned agent networks without the ensemble. When updating, each global Q-network receives the same mini-batch $B$ with size $N_B$. To calculate the target global Q-value, each global Q-network with network parameters $\phi^h$ uses its own target network backup $\bar{\phi}^h$. The loss function for the $h$'th global Q-value network is

$$L(\phi^h) = \frac{1}{N_B} \sum_{(s_t, \boldsymbol{a}_t, s_{t+1}) \in B} (Q_{tot}^{\phi^h}(s_t, \boldsymbol{a}_t) - (r + \gamma \max_{\boldsymbol{a}_{t+1}} Q_{tot}^{\bar{\phi}^h}(s_{t+1}, \boldsymbol{a}_{t+1})))^2. \tag{5}$$

The agent receives the independent gradient flow from each global Q-network to update its policy or individual Q-value network. Dormant neuron rates with different global Q-network ensemble sizes under a replay ratio of 10 are shown in Figure 2(e) and 2(f). As we see, the dormant neuron rate decreases as the ensemble size increases. We hypothesize that, the global Q-network ensemble smooths the gradients for the network updating to reduce the dormant neurons, and Figure 2 (g-h) support our hypothesis as the gradient norm decreases if the ensemble size increases. In addition to the global Q-network ensemble, to maintain the network plasticity under high replay ratios, we incorporate the multiagent Shrink & Perturb strategy (Yang et al., 2024) in Equation (3) and (4) to reset the network parameters to the initial parameters periodically. As in Figure 2 (e-f), this combined ensemble reset operation on the global Q-network further reduces the dormant neuron rate. Moreover, we eliminate the possibility that standard MARL algorithms with a single large global Q-network achieve similar effects to the proposed ensemble reset in Appendix G.

---

**Algorithm 1** Multiagent Reinforcement Learning with Ensemble Reset (EnSet)

---

1: Initialize each agent's policy or individual Q-value network parameters $\theta^1, \theta^2, ..., \theta^N$, an ensemble of $H$ global Q-value network parameters $\phi^1, \phi^2, ..., \phi^H$ and an empty replay buffer $D$. Set target network parameters $\bar{\theta}^i \leftarrow \theta^i$ for $i = 1, 2, ..., N$, and $\bar{\phi}^h \leftarrow \phi^h$ for $h = 1, 2, ..., H$. Set network update interval $T_U$, target network update interval $T_C$, and network reset interval $T_R$.
2: **for** each time step $t$ **do**
3:     Each agent $i$ takes action $a_{i,t} \sim \pi_{\theta^i}(\cdot|o_{i,t})$. Step into state $s_{t+1}$. Receive $r_t$ and observe $o_{i,t+1}$.
4:     Add transition data to the replay buffer: $D \leftarrow D \cup \{(s_t, \boldsymbol{o}_t, \boldsymbol{a}_t, r_t, s_{t+1}, \boldsymbol{o}_{t+1})\}$.
5:     **if** $t \mod T_U = 0$ **then**
6:         **for** each update time $n_{RR}$ from 1 to $N_{RR}$ **do**
7:             Sample a mini-batch $B = \{(s, \boldsymbol{o}, \boldsymbol{a}_t, r, s', \boldsymbol{o}')\}$ from $D$.
8:             Perform the multiagent translation augmentation on global state $s$ to get $\hat{s}$ in sampled $B$.
9:             Update each global Q-network on augmented mini-batch $B$ by

$$L(\phi^h) = \frac{1}{N_B} \sum_{(\hat{s}_t, \boldsymbol{a}_t, \hat{s}_{t+1}) \in B} (Q_{tot}^{\phi^h}(\hat{s}_t, \boldsymbol{a}_t) - (r + \gamma \max_{\boldsymbol{a}_{t+1}} Q_{tot}^{\bar{\phi}^h}(\hat{s}_{t+1}, \boldsymbol{a}_{t+1})))^2.$$

10:             Update each agent's policy or Q-value networks $\theta^1, \theta^2, ..., \theta^N$ on augmented $B$.
11:         **end for**
12:     **end if**
13:     **if** $t \mod T_C = 0$ **then**
14:         Update each target global Q-network $\bar{\phi}^h \leftarrow \phi^h$ and each target agent network $\bar{\theta}^i \leftarrow \theta^i$.
15:     **end if**
16:     **if** $t \mod T_R = 0$ **then**
17:         Shrink & Perturb as Equation (3) and (4) on each global Q-network $\phi^h$ and agent network $\theta^i$.
18:     **end if**
19: **end for**

---

## 3.3 GLOBAL Q-VALUE TRANSLATION INVARIANCE

Beyond the ensemble reset to enable training MARL at high replay ratios, we found that diversifying the replay experience via multiagent intrinsic properties is also effective to enhance the sample efficiency. Learning at a high replay ratio would make the agents incur a risk of overfitting to earlier experiences, negatively affecting the rest of the learning process (Nikishin et al., 2022). Therefore, the multiagent experience augmentation techniques could be naturally incorporated into the high-replay-ratio training setting of MARL. In this paper, we explore the multiagent translation invariance

(Vasile et al., 2015) of the global Q-function to augment the state input of the global Q-network. For the multiagent translation invariance, we have the following definition (Vasile et al., 2015).

**Definition 2** (Multiagent Translation Invariance). In a $D$-dimensional coordinate system, a function $f : S \rightarrow Y$, where the global state $s \in S$ contains features of multiple agents and units as $s = (x_1, \cdots, x_N, x_o)$, $x_i = (c_i, e_i)$ includes the agent $i$'s coordinate features $c_i \in \mathbb{R}^D$ as well as agent $i$'s extra unit features $e_i$, and $x_o$ represents other system features such as joint actions if $f$ needs, is said to be translation invariant if for all $z \in \mathbb{R}^D$ the following condition holds:

$$f((c_1 + z, e_1), \cdots, (c_N + z, e_N), x_o) = f(x_1, \cdots, x_N, x_o), \tag{6}$$

where the translation variable $z$ in each coordinate dimension is uniformly sampled from $[-\alpha_z, \alpha_z]$. When $f$ becomes the global Q-value function $Q_{tot}$, we are inspired to apply the multiagent translation invariance into the input global state $s$ of the global Q-network as

$$Q_{tot}((c_1 + z, e_1), \cdots, (c_N + z, e_N), x_o) = Q_{tot}(x_1, \cdots, x_N, x_o). \tag{7}$$

This invariance property also applies to other environmental units in multiagent systems if they have coordinate features. Although the multiagent translation invariance is a natural data augmentation operation for the global Q-value function, few works utilize this property in the MARL domain.

We give the algorithmic descriptions of EnSet in Algorithm 1. EnSet follows the standard workflow of off-policy MARL algorithms. Specifically, in Lines 1, EnSet employs an ensemble of $H$ global Q-value networks. In Line 8, EnSet transforms global state $s$ to $\hat{s}$ in mini-batch $B$ with the multiagent translation. Then, in Line 9, each global Q-network is updated with a temporal difference error calculated with $\hat{s}$. Finally, in Line 17, given the network reset interval $T_R$ time steps, EnSet performs Shrink & Perturb to inject plasticity into both the global Q-network networks and agent networks to keep learning ability. Next, we evaluate the efficacy of EnSet with extensive experiments.

## 4 EXPERIMENTS

In this section, we conduct extensive experiments to validate EnSet. First, we integrate EnSet into classical MARL algorithms such as QMIX (Rashid et al., 2018), QPLEX (Wang et al., 2021), and ATM (Yang et al., 2022) in the well-known StarCraft Multi-Agent Challenge (SMAC) (Samvelyan et al., 2019) of a discrete action space (Section 4.1). Second, we compare EnSet with other multiagent network reset techniques in the high-replay-ratio setting (Section 4.2). Third, we conduct the ablation study to validate each component of EnSet in Section 4.3. Fourth, we further validate EnSet in the classical multiagent particle (MPE) environment (Lowe et al., 2017) of the continuous action space by combining it with MADDPG (Lowe et al., 2017) and FACMAC (Peng et al., 2021) in Section 4.4. Fifth, experiments of EnSet with finetuned QMIX in SMACv2 (Ellis et al., 2023) are given in Section 4.5. Finally, we present the replay ratio scaling experiments in Section 4.6.

### 4.1 STARCRAFT II MULTI-AGENT CHALLENGE

First, we experiment in the well-known StarCraft II Multi-Agent Challenge (SMAC) (Samvelyan et al., 2019), a widely adopted testbed consisting of decentralized micromanagement tasks for evaluating MARL approaches. We train multiple agents to control allied units respectively to beat the enemies, while a built-in handcrafted AI controls these enemy units. The SMAC environment is with discrete action space, and the version of StarCraft II is 4.6.2. We implement EnSet based on the pymarl framework (Samvelyan et al., 2019) with default training and evaluation configurations.

In SMAC, we evaluate on 8 tasks, i.e., 2s3z, 10m_vs_11m, 3s5z, 5m_vs_6m, MMM2, 3s5z_vs_3s6z, 6h_vs_8z, and corridor. These tasks include homogeneous and heterogeneous multiagent scenarios, as well as symmetrical and asymmetrical multiagent scenarios for a comprehensive evaluation. For example, on the task of 2s3z, each side has 2 Stalkers and 3 Zealots units, where the multiagent system is heterogeneous and symmetrical. More descriptions of these tasks are given in Appendix C.

First, we show that EnSet is able to boost the performance of popular MARL algorithms under high replay ratios. We integrate EnSet into QMIX, QPLEX, and ATM with a replay ratio of 10, where 10 updates are executed after one episode. For EnSet, the ensemble size $H$ of the global Q-network is set at 5. The coordinate translation variable $\alpha_z$ in Equation (6) is set at 0.2. The hyperparameters of

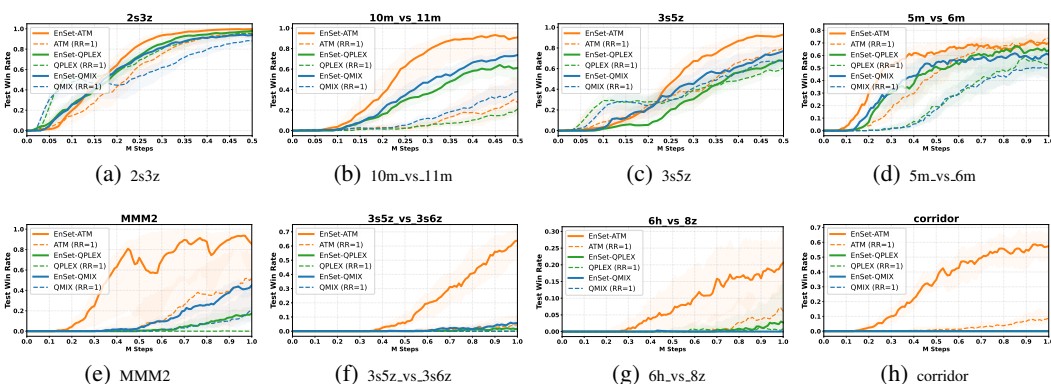

Figure 3: Benchmark experiments of EnSet for boosting MARL algorithms in SMAC within limited environment interactions. Enset-QMIX, Enset-QPLEX, and EnSet-ATM are trained with a replay ratio of 10, while QMIX, QPLEX, and ATM are trained with a default replay ratio value of 1.

multiagent Shrink & Perturb follow MARR (Yang et al., 2024). All these special hyperparameters of EnSet are summarized in Appendix B and kept the same across methods and tasks. We run all methods over 6 independent training runs with different random seeds.

The experimental results are shown in Figure 3, including the median performance as well as the shaded 25-75% percentiles. As we see, EnSet significantly boosts QMIX, QPLEX, and ATM. For the easy scenarios, including 2s3z, 10m_vs_11m, and 3s5z, the number of environment interactions is set at 0.5 million. With such a small interaction budget, EnSet successfully speeds up the standard MARL algorithms especially on 10m_vs_11m, demonstrating superior sample efficiency. For the remaining scenarios, we set the number of environment interactions at 1 million steps. EnSet also substantially speeds up the learning of each MARL algorithm, while the standard ones with a replay ratio of 1 learn slowly. Specifically, in the super hard scenarios such as 3s5z_vs_3s6z and corridor, EnSet successfully improves the performance of ATM by a large margin within only 1 million steps. These results in SMAC show that EnSet is efficient in boosting MARL algorithms at a high replay ratio to achieve sample efficiency given the same environment interactions.

Additionally, results of EnSet with different global Q-network ensemble sizes are reported in Appendix E. Results of EnSet with a standard replay ratio of 1 are given in Appendix F. The comparison of EnSet with standard MARL algorithms having more environment steps is shown in Appendix H.

## 4.2 COMPARING WITH MARL NETWORK RESET METHODS

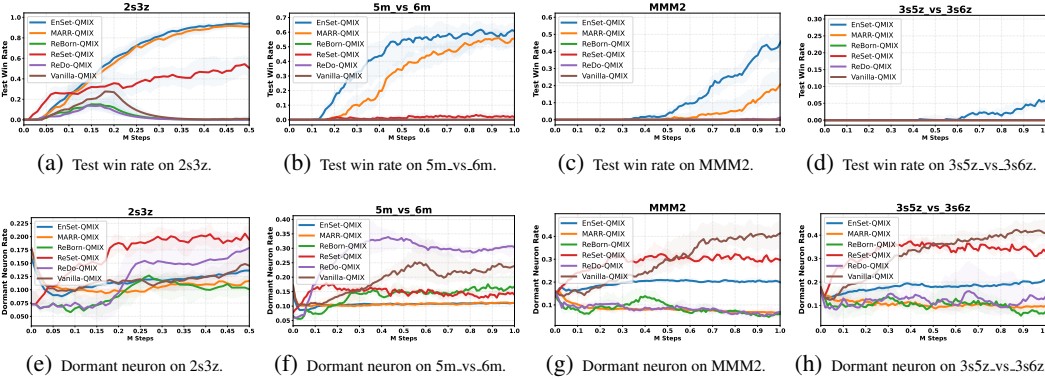

Figure 4: Benchmarking different MARL network reset methods in SMAC. Replay ratio is set at 10.

Second, we compare EnSet with other MARL network reset methods such as MARR (Yang et al., 2024), ReBorn (Qin et al., 2024), ReSet (Nikishin et al., 2022), and ReDo (Sokar et al., 2023) on the basis of QMIX. MARR (Yang et al., 2024) introduces the Shrink & Perturb strategy into MARL to reset network parameters periodically. ReBorn (Qin et al., 2024) transfers the weights from over-active neurons to dormant neurons to prevent both the overactive and dormant neurons in MARL networks. ReSet (Nikishin et al., 2022) addresses early agent experience bias by periodically resetting the last layer of the neural network to avoid overfitting. ReDo (Sokar et al., 2023) periodically reinitializes the input weights of dormant neurons and zeros the output weights of dormant neurons. We also present the vanilla QMIX, which is directly trained at a replay ratio of 10, as a comparison. The corresponding performance is plotted in Figure 4. At the same time, we also give the dormant neuron rates in the global Q-network to see how these methods work. It is clear that EnSet maintains the dormant neuron rates at a low level and achieves the best performance among baselines. Interestingly, although MARR, ReDo, and ReBorn have lower dormant neuron rates than EnSet in MMM2 and 3s5z_vs_3s6z, they perform worse than EnSet. This indicates that, while a high dormant neuron rate prevents MARL from stabilizing learning, forcing a much lower dormant neuron rate may constrain network parameters and hurt network representation ability. Moreover, an in-depth comparison between EnSet and MARR is provided in Appendix D for reference.

## 4.3 ABLATION STUDY OF ENSET

Here, we also conduct the ablation study to validate each component of EnSet under high replay ratios in SMAC and MPE. There are three key techniques in EnSet such as the global Q-network ensemble, the multiagent Shrink & Perturb strategy, as well as the multiagent state translation. Results of the ablation study on these components are shown in Figure 5.

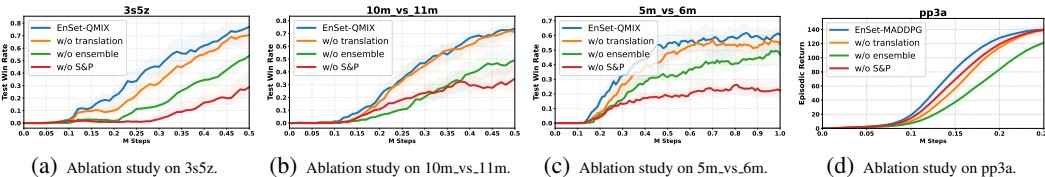

(a) Ablation study on 3s5z.  (b) Ablation study on 10m_vs_11m.  (c) Ablation study on 5m_vs_6m.  (d) Ablation study on pp3a.

Figure 5: The ablation study of EnSet-QMIX in SMAC and EnSet-MADDPG in MPE. Replay ratio is set at 10 in SMAC and 25 in MPE. 'w/o ensemble' means the number of global Q-networks in EnSet is 1. 'w/o translation' means the multiagent translation invariance is not applied in the global states. 'w/o S&P' means the multiagent Shrink & Perturb strategy is not used in EnSet.

We see that the global Q-network ensemble and multiagent Shrink & Perturb in ensemble reset largely affects performance, implying that tackling dormant neurons is the key to keeping learning at high replay ratios (Yang et al., 2024). Specifically, the ensemble contributes the most to the performance in pp3a, further highlighting its importance. Meanwhile, the multiagent state translation slightly improves performance consistently during the learning process in all these tested scenarios.

## 4.4 MULTIAGENT PARTICLE ENVIRONMENT

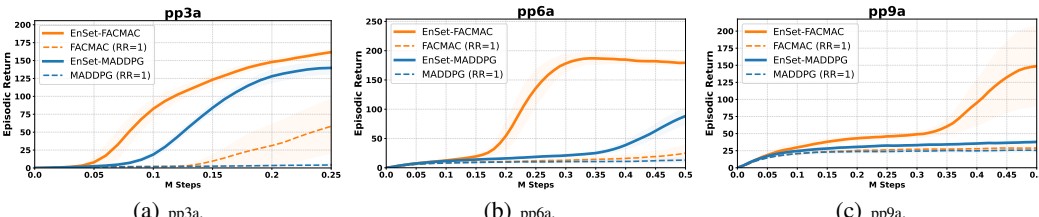

(a) pp3a.  (b) pp6a.  (c) pp9a.

Figure 6: Benchmark experiments of EnSet for boosting MARL algorithms in the MPE environment. Enset-MADDPG and EnSet-FACMAC are trained with a replay ratio of 25, while the standard MADDPG and FACMAC algorithms are trained with a default replay ratio value of 1.

We further evaluate EnSet in the Multiagent Particle Environment (MPE) (Lowe et al., 2017), which features a continuous action space. Following established benchmarks (Peng et al., 2021), we adopt a set of predator-prey tasks where multiple slower cooperative agents must capture faster prey in a continuous two-dimensional toroidal space with obstacle landmarks. More details are provided in Appendix C. We experiment on 3 tasks with different agent numbers, i.e., pp3a where 3 agents catch 1 prey, pp6a where 6 agents catch 2 preys, and pp9a where 9 agents catch 3 preys. All training configurations strictly follow Peng et al. (2021) for fair comparison. Hyperparameters of EnSet in MPE are the same as in SMAC and listed in Appendix B. Next, we test EnSet on these varying scenarios by combining it with MADDPG and FACMAC with a replay ratio of 25. Results in MPE are shown in Figure 6, which are averaged over 6 independent runs with a 95% confidence interval. For the easy pp3a, the environment interactions are 0.25 million steps. For pp6a and pp9a, the number of environment steps is 0.5 million. Impressively, EnSet boosts the learning of both MADDPG and FACMAC, indicating EnSet is general for off-policy MARL algorithms.

## 4.5 THE SMACv2 ENVIRONMENT

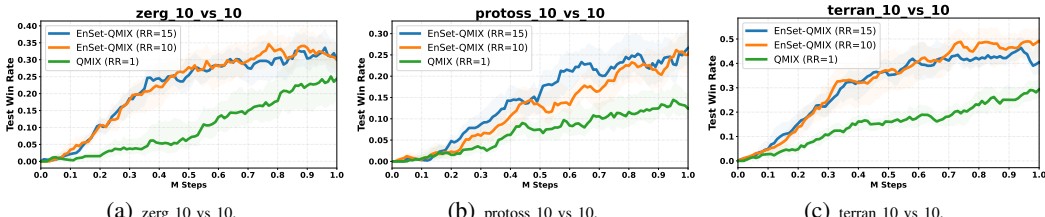

(a) zerg_10_vs_10.    (b) protoss_10_vs_10.    (c) terran_10_vs_10.

Figure 7: Benchmark experiments of EnSet for boosting MARL in SMACv2. Enset-QMIX is trained with a replay ratio of 10 or 15, while QMIX is trained with a default replay ratio value of 1.

In this section, we evaluate EnSet in SMACv2 (Ellis et al., 2023), which is established to enforce sufficient stochasticity and meaningful partial observability for benchmarking MARL algorithms. We conduct the experiments with EnSet on 3 scenarios, including zerg_10_vs_10, protoss_10_vs_10, and terran_10_vs_10. More details of these SMACv2 tasks refer to Appendix C. In SMACv2, we build EnSet on top of pymarl2 (Hu et al., 2023), which introduces various code-level optimizations for MARL, such as $N$-step returns, large batch size, Adam optimizer, and so on. We follow the default hyperparameters of pymarl2, where the sampling environment number is optimized to 8. The hyperparameters of EnSet are the same as in SMAC and MPE. We experimented with EnSet-QMIX using two replay ratios of 10 and 15. The results are shown in Figure 7, including the median performance as well as the shaded 25-75% percentiles. We see that, under both high replay ratios, EnSet successfully boosts QMIX in these challenging SMACv2 tasks.

## 4.6 DIFFERENT REPLAY RATIOS

In this section, we demonstrate how different replay ratios affect EnSet's performance. We experiment with EnSet using different replay ratios, such as 10, 25, and 50, for both QMIX on 5m_vs_6m and MADDPG on pp3a. The performance metrics and dormant neuron rates are shown in Figure 8. We see that high replay ratios lead to severe dormant neuron rates, and EnSet helps greatly reduce dormant neurons. On the other hand, the optimal replay ratio depends on environments and methods. A replay ratio of 10 is better than 25 and 50 in 5m_vs_6m, while replay ratios of 25 and 50 are better than 10 in pp3a. Nevertheless, EnSet always stabilizes MARL at different high replay ratios.

## 5 CONCLUSION

In this paper, we propose EnSet to stabilize MARL training at high replay ratios for sample efficiency. EnSet consists of two key innovations: the global Q-network ensemble reset and global Q-value translation invariance. First, we found that the dormant neuron phenomenon becomes severe when the replay ratio is high, and the global Q-network ensemble with network parameter reset mitigates the high dormant neuron rate issue to stabilize the learning process of MARL. Second, we

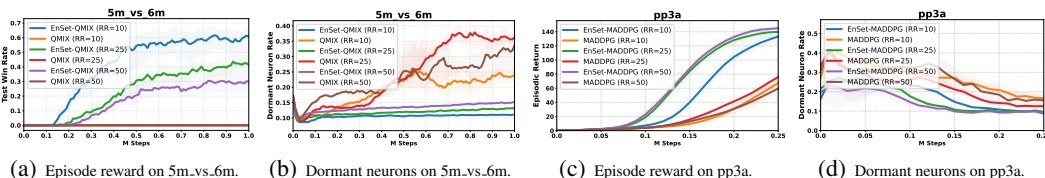

(a) Episode reward on 5m_vs_6m.  (b) Dormant neurons on 5m_vs_6m.  (c) Episode reward on pp3a.  (d) Dormant neurons on pp3a.

Figure 8: Experimental results of EnSet under different replay ratios.

introduce the multiagent translation invariance into the global state to generate more replay experience for the global Q-networks. Extensive experiments in SMAC, MPE, and SMACv2 show that EnSet speeds up the learning of MARL to a new degree at the high-replay-ratio setting.

For future work, dynamically adjusting the replay ratio throughout learning is interesting. Second, integrating EnSet into on-policy MARL algorithms (e.g., MAPPO, COMA) is worth exploring. Third, a deeper and theoretical analysis of how the dormant neuron phenomenon in MARL arises and proceeds during training is also important.

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

# A    MORE EXPERIMENTAL RESULTS OF DORMANT NEURONS IN HIGH REPLAY RATIO SETTING

Here, we show more results of dormant neurons in the high-replay-ratio setting in Figure 9. We observe that, when the replay ratio is either 5 or 10, the dormant neuron rate in the global Q-network is higher than when the replay ratio is 1. Furthermore, when the replay ratio reaches 10, the learning cripples in all the tested cases. This necessitates EnSet, which stabilizes MARL at high replay ratios by tackling dormant neurons in the global Q-network. At the same time, the dormant neuron rate in the agent network maintains a low level even when the replay ratio increases.

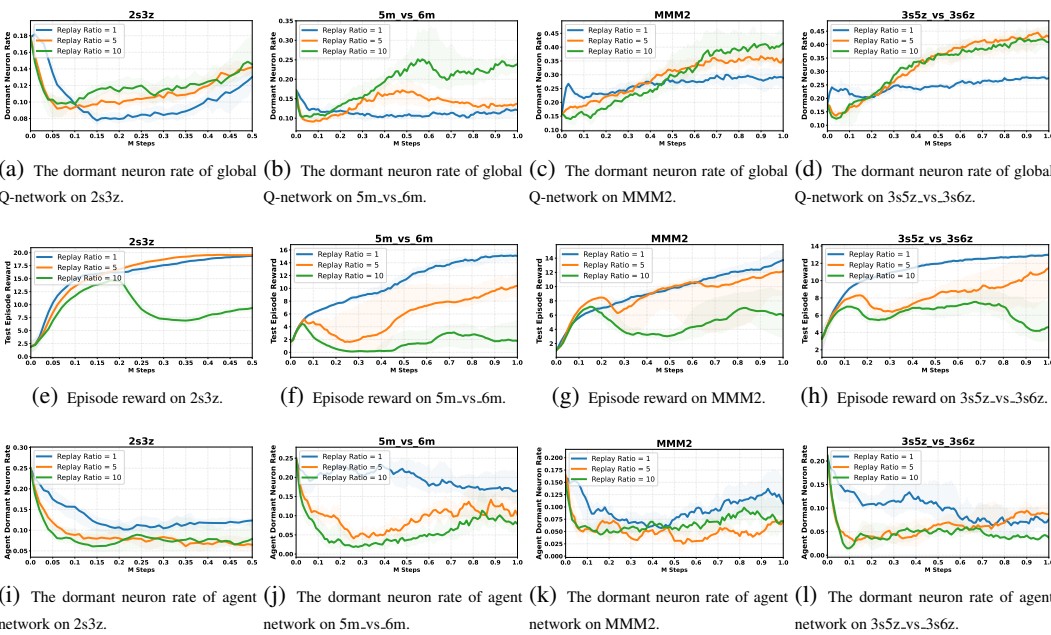

(a) The dormant neuron rate of global Q-network on 2s3z.

(b) The dormant neuron rate of global Q-network on 5m_vs_6m.

(c) The dormant neuron rate of global Q-network on MMM2.

(d) The dormant neuron rate of global Q-network on 3s5z_vs_3s6z.

(e) Episode reward on 2s3z.

(f) Episode reward on 5m_vs_6m.

(g) Episode reward on MMM2.

(h) Episode reward on 3s5z_vs_3s6z.

(i) The dormant neuron rate of agent network on 2s3z.

(j) The dormant neuron rate of agent network on 5m_vs_6m.

(k) The dormant neuron rate of agent network on MMM2.

(l) The dormant neuron rate of agent network on 3s5z_vs_3s6z.

Figure 9: The dormant neuron phenomenon in the global Q-network is exacerbated when improving replay ratios. Higher replay ratios cause learning instability. When the replay ratio is 10, the learning collapses with lots of dormant neurons being inactive in the global Q-network.

# B    HYPERPARAMETERS OF ENSET

All special hyperparameters of EnSet are the same across algorithms, tasks, and environments in this paper. The global Q-network ensemble size of EnSet $H$ is set at 5. The translation variable $\alpha_z$ for multiagent translation in Equation (6) is set at 0.2. The hyperparameters of multiagent Shrink & Perturb follow MARR (Yang et al., 2024), where the interpolation factor $\alpha_r$ is set at 0.8 and the reset interval $T_R$ is set at 2000. For the agent observation enhancement, we follow MARR to enable random amplitude scaling (Laskin et al., 2020) on agent observation features in EnSet, where the scale variable is sampled from a uniform distribution over a range $[0.8, 1.2]$. The special hyperparameters of EnSet are summarized in Table 1. Other hyperparameters except the replay ratio follow the standard MARL algorithms.

Table 1: Summary of special hyperparameters of EnSet.

| EnSet hyperparameters | Component | Value |
|---|---|---|
| global Q-network ensemble size of EnSet $H$ | Ensemble Reset | 5 |
| interpolation factor $\alpha_r$ | Ensemble Reset | 0.8 |
| reset interval $T_R$ | Ensemble Reset | 2000 |
| translation variable $\alpha_z$ | Multiagent Translation | 0.2 |
| scale variable range | Random Amplitude Scaling | [0.8, 1.2] |

# C   SUMMARY OF EXPERIMENTAL TASKS AND COMPUTING RESOURCES

Here, we briefly summarize the tasks in SMAC, MPE, and SMACv2 environments in Table 2.

There are 8 tasks in SMAC. For map 2s3z, each side has 2 Stalkers and 3 Zealots. For map 3s5z, both sides have 3 Stalkers and 5 Zealots. In the map of 10m_vs_11m, there are 10 allied marines against 11 marine enemies. In 5m_vs_6m, there are 5 allied marines against 6 marine enemies. In MMM2, there are 1 Medivac, 2 Marauders, and 7 Marines against 1 Medivac, 3 Marauders, and 8 Marines. In 3s5z_vs_3s6z, there are 3 Stalkers and 5 Zealots against 3 Stalkers and 6 Zealots. In corridor, 6 allied Zealots are against 24 Zerglings. In 6h_vs_8z, there are 6 Hydralisks against 8 Zealots. At each episode, a group of allied units is going to fight against the enemy units.

In MPE, we use a set of predator-prey tasks, where multiple slower cooperative agents must capture faster prey in a continuous two-dimensional toroidal space with obstacle landmarks. If one agent collides with the prey while at least another one is close enough, a team reward of +10 is given. However, if only one agent collides with the prey without any other agent being close enough, a negative team reward of -1 is given. Otherwise, no reward is provided. In this task, each agent can observe its own position and velocity, the relative positions of other agents, the relative position and velocity of preys, and the relative positions of landmarks. Additionally, each agent has a view radius, which restricts the agents from receiving information about other entities (including all landmarks, other agents, and preys) that are outside the view.

In SMACv2, task scenarios are procedurally generated and require agents to generalize to previously unseen settings during evaluation. We experiment with three challenging tasks, including zerg_10_vs_10, protoss_10_vs_10, and terran_10_vs_10. In zerg_10_vs_10, there are 10 allied zerg units randomly generated to fight against 10 zerg enemy units, which are also randomly generated. In protoss_10_vs_10, 10 randomly generated protoss agents are controlled to fight against 10 protoss enemies. In terran_10_vs_10, 10 allied terran units fight against 10 terran enemy units.

Table 2: Summary of experimental tasks in SMAC, MPE, and SMACv2.

| SMAC | | |
|---|---|---|
| **Task Name** | **Allied Units** | **Enemy Units** |
| 2s3z | 2 Stalkers, 3 Zealots | 2 Stalkers, 3 Zealots |
| 10m_vs_11m | 10 Marines | 11 Marines |
| 3s5z | 3 Stalkers, 5 Zealots | 3 Stalkers, 5 Zealots |
| 5m_vs_6m | 5 Marines | 6 Marines |
| MMM2 | 1 Medivac, 2 Marauders, 7 Marines | 1 Medivac, 3 Marauders, 8 Marines |
| 3s5z_vs_3s6z | 3 Stalkers, 5 Zealots | 3 Stalkers, 6 Zealots |
| 6h_vs_8z | 6 Hydralisks | 8 Zealots |
| corridor | 6 Zealots | 24 Zerglings |
| **MPE** | | |
| **Task Name** | **Agents** | **Preys** |
| 3a | 3 Agents | 1 Prey |
| 6a | 6 Agents | 2 Preys |
| 9a | 9 Agents | 3 Preys |
| **SMACv2** | | |
| **Task Name** | **Allied Units** | **Enemy Units** |
| zerg_10_vs_10 | 10 Randomly Zerg Units | 10 Randomly Zerg Units |
| protoss_10_vs_10 | 10 Randomly Protoss Units | 10 Randomly Protoss Units |
| terran_10_vs_10 | 10 Randomly Terran Units | 10 Randomly Terran Units |

For the computing resources, the experiments are conducted with the NVIDIA GPUs. The version of PyTorch is 2.6.0. The operating system is Ubuntu. For a single instance of all the algorithms except ATM and EnSet-ATM, the amount of GPU memory is small, which is usually less than 1GB. For a single instance of ATM and EnSet-ATM based on transformer layers, the maximum GPU memory is about 24GB on corridor, where a total of up to 30 units exist in this map.

Table 3: Summary of differences of EnSet and MARR in experiments.

| Method | EnSet | MARR |
|---|---|---|
| Multiagent Shrink & Perturb | Yes | Yes |
| Global Q-network ensemble | Yes | No |
| Agent observation augmentation | Random amplitude scaling | Random amplitude scaling |
| Global state augmentation | Multiagent translation | Random amplitude scaling |
| Number of Environments in SMAC | 1 | 8 |
| Replay ratio in SMAC | 10 | 50 |
| Number of Environments in MPE | 1 | 4 |
| Replay ratio in MPE | 25 | 25 |

## D    MORE COMPARISONS OF ENSET WITH MARR

In this section, we conduct an in-depth comparison with MARR with its official default settings, such as the number of environments for sampling. We first list the key differences between EnSet and MARR in Table 3. As we see, MARR employs parallel environments for scaling the replay ratio while EnSet uses the default series environment in both SMAC and MPE. The interpolation factor $\alpha_r$ of 0.8 in multiagent Shrink & Perturb is the same in both EnSet and MARR, and the reset interval $T_R$ is set at 2000 for both methods. EnSet integrates the global Q-network ensemble, while MARR does not consider it. The ensemble size of global Q-networks in EnSet is set at 5. For the local observation, we augment EnSet's agent observation with the random amplitude scaling by following MARR with the same hyperparameters. In EnSet, we apply the multiagent translation to diversify the global state $s$, which implies the global Q-value invariance. In contrast, the random amplitude scaling in MARR causes biases in the estimation of global Q-values. For example, increasing the allied unit health would increase the global Q-value, which departs from its true value to an overestimation. EnSet and MARR also have different replay ratios in SMAC, where EnSet uses 10 and MARR uses 50. The replay ratio is set at 25 for both EnSet and MARR in MPE.

Next, we compare EnSet and MARR with their default settings at the same environment interaction steps. We integrate EnSet and MARR into the best standard algorithms (i.e., ATM in SMAC and FACMAC in MPE). The results are shown in Figure 10. In both simple scenarios, 10m_vs_11m and pp3a, as well as difficult scenarios, 3s5z_vs_3s6z and pp6a, EnSet outperforms MARR within the same environment steps, regardless of different settings such as parallel/series environments and replay ratios. This indicates that EnSet achieves a higher sample efficiency than MARR.

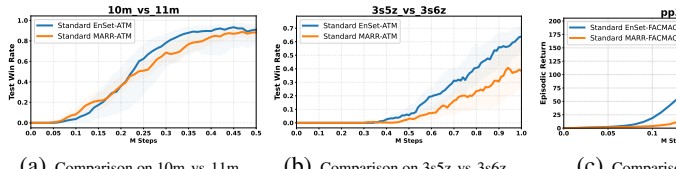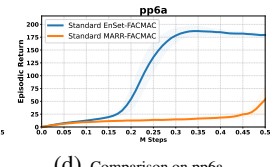

(a) Comparison on 10m_vs_11m.  (b) Comparison on 3s5z_vs_3s6z.  (c) Comparison on pp3a.  (d) Comparison on pp6a.

Figure 10: The comparison of standard EnSet and MARR in SMAC as well as MPE.

## E    DIFFERENT GLOBAL Q-NETWORK ENSEMBLE SIZES

In this section, we study how the ensemble size of the global Q-network affects the performance of EnSet. In Figure 11, the ensemble size $H$ of 1 performs worst in these cases, while $H$ of 5 consistently performs well in both SMAC and MPE. At the same time, $H$ of 10 also achieves superior performance in 3s5z and pp3a, further proving ensemble's effectiveness at high replay ratios.

## F    ENSET WITH A STANDARD REPLAY RATIO OF 1

Although EnSet is specially designed for the high-replay-ratio training setting, we also wonder about the performance of EnSet under the standard replay ratio of 1. Therefore, we provide the results in

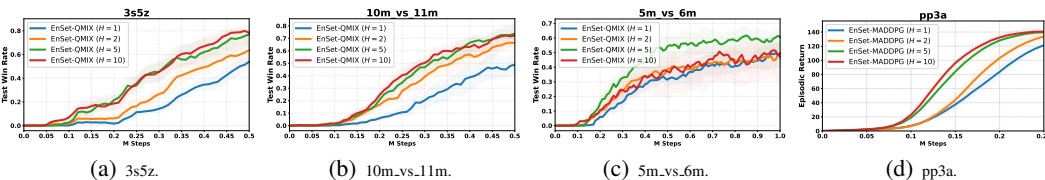

Figure 11: Experimental evaluation of the performance of different ensemble sizes in EnSet.

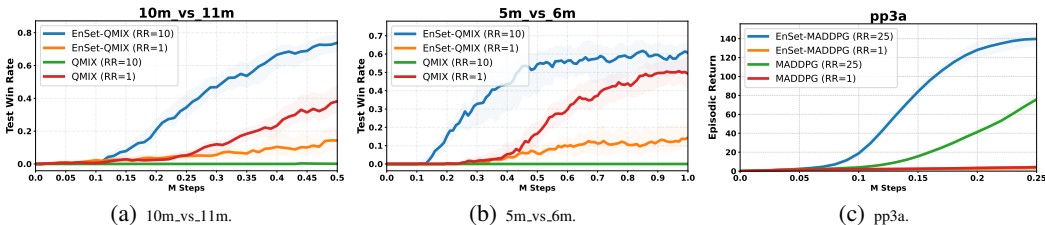

Figure 12: Experimental evaluation of the performance of EnSet with a replay ratio of 1.

Figure 12. EnSet with a replay ratio of 1 performs worse than the standard MARL algorithms with a replay ratio of 1, which is expected as network reset is employed to forget past experience while no severe dormant neuron phenomenon exists. However, both the standard QMIX and MADDPG fail to learn at high replay ratios. On the other hand, EnSet at high replay ratios achieves a significant improvement over standard MARL algorithms with a replay ratio of 1. These results confirm that EnSet is specifically designed to address the challenge of training MARL at high replay ratios.

## G  COMPARISON OF ENSEMBLE WITH A LARGE GLOBAL Q-NETWORK

In this section, we examine whether the standard MARL algorithms with a single large global Q-network achieve similar results to EnSet-based algorithms with the ensemble reset. We refer to the default model size of the global Q-network in the standard MARL algorithms as the standard size. At the same time, we set the model size of the global Q-network in the standard MARL algorithms to the same size as EnSet's global Q-network ensemble. We also enlarge the model size of the global Q-network to be twice the size of EnSet's global Q-network ensemble. Results of MARL algorithms with standard model size, model size as EnSet's global Q-network ensemble, and model size twice as EnSet's global Q-network ensemble are in Figure 13. The standard MARL algorithms with various model sizes fail to learn at high replay ratios, showing the ensemble reset's effectiveness.

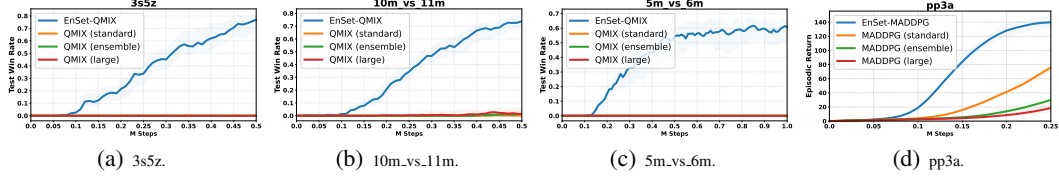

Figure 13: The comparison results of EnSet and the standard MARL algorithms with a large global Q-network. 'ensemble' indicates the model size equals the size of EnSet's ensembled global Q-networks. 'large' indicates the model size is twice the size of EnSet's ensembled global Q-networks.

## H  COMPARING STANDARD MARL WITH MORE ENVIRONMENT STEPS

In this section, we show that EnSet outperforms the advanced MARL algorithms with a standard replay ratio of 1, even given more environment interaction steps. To make a practical comparison,

we experiment on the most challenging tasks, including 3s5z_vs_3s6z and corridor in SMAC and pp9a in MPE. At the same time, we use advanced MARL algorithms such as ATM and FACMAC as the tested algorithms, which show superior performance in SMAC and MPE, respectively. The comparison results are plotted in Figure 14. It shows that, even though the number of environment interaction steps of the standard ATM is twice that of EnSet-ATM, EnSet-ATM's performance is still better. A similar result is also for the standard FACMAC and EnSet-FACMAC. In pp9a, while the standard FACMAC's environmental steps are four times of EnSet-FACMAC, its final policy performs worse than EnSet-FACMAC, further showing EnSet's impressive sample efficiency.

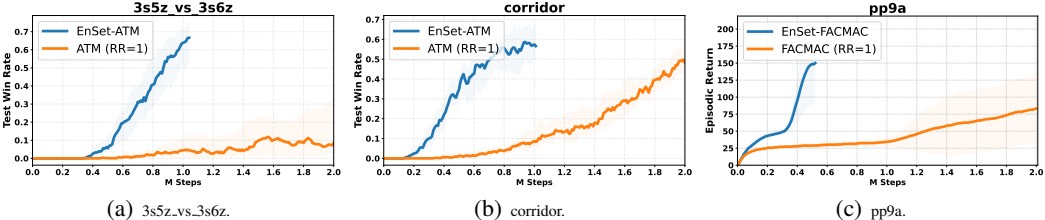

(a) 3s5z_vs_3s6z.  (b) corridor.  (c) pp9a.

Figure 14: The comparison of EnSet with standard MARL algorithms with more environment steps.

## I  INTRODUCING ENSET TO OTHER NETWORK RESET TECHNIQUES

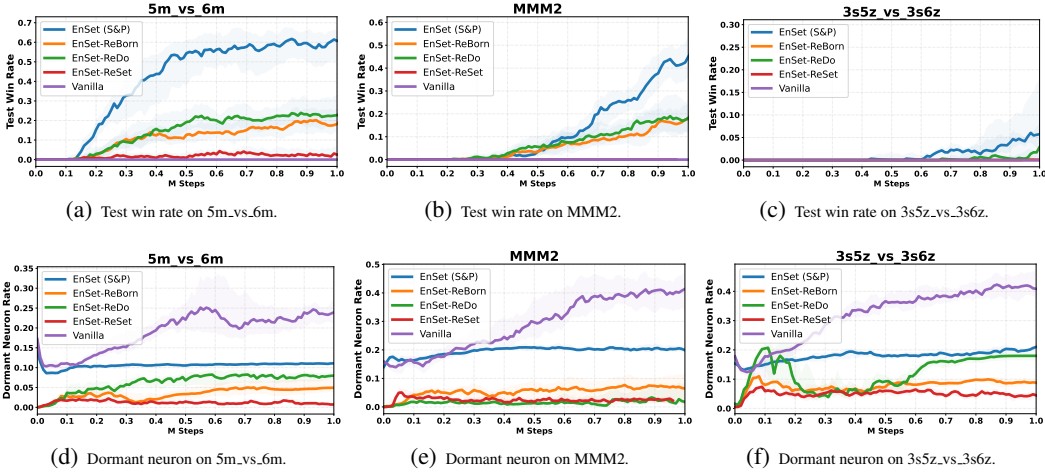

(a) Test win rate on 5m_vs_6m.  (b) Test win rate on MMM2.  (c) Test win rate on 3s5z_vs_3s6z.

(d) Dormant neuron on 5m_vs_6m.  (e) Dormant neuron on MMM2.  (f) Dormant neuron on 3s5z_vs_3s6z.

Figure 15: Introducing EnSet to other network reset techniques by replacing the Shrink & Perturb (denoted as S&P) with ReBorn, ReDo, and ReSet in SMAC. Replay ratio is set at 10. The base algorithm is QMIX, and Vanilla indicates QMIX with a replay ratio of 10.

In this section, we replace the Shrink & Perturb in the standard EnSet with other MARL network reset techniques, such as ReBorn, ReDo, and ReSet. Results are shown in Figure 15. First, although all methods mitigate the severe dormant neurons in the vanilla QMIX, EnSet is the most compatible with Shrink & Perturb, achieving the best performance among all these methods. Second, although EnSet-based ReSet has the lowest dormant neuron rates (almost to 0) in all maps, it performs poorly, further confirming that forcing a much lower dormant neuron rate may constrain network parameters and hurt network representation ability. These results show that the Shrink & Perturb is compatible with EnSet best among these reset techniques, which may benefit from its soft-reset mechanism.

## J  BROADER IMPACTS

MARL is a practical paradigm that models real-world scenarios. The proposed EnSet stabilizes the learning of existing off-policy MARL algorithms at high replay ratios to learn satisfactory polices

within limited environment interaction steps. However, when applied to real-world tasks, EnSet-based MARL algorithms still need some exploration steps to learn, which may lead to unsafe situations. On the other hand, the findings that high replay ratios exacerbate dormant neurons in the global Q-network may inspire the development of single-agent reinforcement learning algorithms.

## K    LIMITATIONS

Our study may have limitations under extensive consideration. One main limitation of EnSet is that it is not suitable for the on-policy MARL algorithms, as it utilizes the replay buffer with a high replay ratio. Second, the global Q-network ensemble in EnSet increases the computation cost in the training stage, therefore increasing the running time of instances.

