# OpenReview forum: "Boosting Multiagent Reinforcement Learning at High Replay Ratios with Ensemble Reset"
_ICLR.cc/2026/Conference — ICLR 2026 Conference Withdrawn Submission_

### Official Review · Reviewer_XSmN · 2025-10-16

**Soundness:** 3
**Presentation:** 2
**Contribution:** 2
**Rating:** 2
**Confidence:** 3

**Summary:**

The proposed paper introduces an algorithm named EnSet to improve sample efficiency in multi-agent reinforcement learning environments. To achieve this, EnSet employs an ensemble of global Q-value networks and periodically performs a reset operation to reduce dormant neurons. Furthermore, it utilizes translation invariance to prevent overfitting of the global Q-value function.

**Strengths:**

* The paper is easy to understand.
* EnSet can be applied to various algorithms and demonstrates improved sample efficiency.
* The effectiveness of EnSet is verified across diverse environments.

**Weaknesses:**

* The paper attributes improved sample efficiency to the combination of ensemble, reset, and multi-agent translation invariance. However, given that the reset mechanism is adopted from prior work [1] and the empirical impact of the multi-agent translation invariance appears marginal in the results, the primary contribution of the paper seems reduced to the mere application of a standard ensemble technique.

* While achieving strong performance with fewer environment steps is intriguing, this approach, coupled with the ensemble method and a high replay ratio, likely incurs a substantial computational overhead (e.g., training time). A thorough analysis quantifying the computational cost vs. performance gain trade-off is currently missing and required for a complete evaluation.

* The overall clarity of the paper's narrative is inconsistent in specific sections: (1) The purpose and relevance of the content related to Equation 1 in Section 2.2 are ambiguous, particularly as the concept is not subsequently utilized, suggesting it may be superfluous. (2) The third paragraph of Section 4.1 requires significant revision for structural clarity and flow.

* The periodic 'Shrink & Perturb' operation is not clearly defined. It is uncertain whether this mechanism is applied to each individual network ($\phi^{h}$ and $\theta^{i}$) within the ensemble or to the ensemble as a whole($\phi^{1},\cdots,\phi^{H}$ and $\theta^{1},\cdots,\theta^{N}$), which hinders a complete understanding of the algorithm's functional mechanism.

* The experimental results lack justification for using different evaluation metrics (median with 25-75% percentiles and average with 95% confidence interval) across various experiments. Furthermore, the excessive smoothing applied to several plots significantly diminishes the graphical fidelity and reliability of the presented data.

* A comprehensive table summarizing all crucial experimental setup parameters and algorithmic hyperparameters would greatly enhance the reproducibility and clarity of the work.

&nbsp;

[1] Yang, Y., Chen, G., & Heng, P. A. (2024, July). Sample-efficient multiagent reinforcement learning with reset replay. In Forty-first international conference on machine learning.

**Questions:**

* Figures 2 and 4 suggest that merely reducing dormant neurons is insufficient to guarantee performance improvement. Could the authors elaborate on why this occurs, and, in light of these specific control results, what is the mechanism by which EnSet successfully translates reduced dormant neurons into superior performance?

* Considering the goal of mitigating dormant neurons, the 'redo' [2] mechanism has been explored in related literature as an alternative to 'reset'. Could the authors provide experimental results showing the performance of the proposed algorithm when the 'Shrink & Perturb' component is replaced solely with the 'Redo' technique while maintaining other components?

* Could the authors quantify the difference in computational cost (e.g., wall-clock time, GPU/CPU utilization) between the baseline algorithm and the proposed algorithm?

* Previous research [3] in single-agent RL utilizes ensemble methods and reset mechanisms, demonstrating improved sample efficiency as the replay ratio increases, which feels similar to EnSet. What is the authors' opinion on comparing EnSet with this method?

&nbsp;

[2] Sokar, G., Agarwal, R., Castro, P. S., & Evci, U. (2023, July). The dormant neuron phenomenon in deep reinforcement learning. In International Conference on Machine Learning (pp. 32145-32168). PMLR.

[3] Kim, W., Shin, Y., Park, J., & Sung, Y. (2023). Sample-efficient and safe deep reinforcement learning via reset deep ensemble agents. Advances in neural information processing systems, 36, 53239-53260.

---

> ### Author Response · Authors · 2025-11-24
> **Rebuttal part 1 (out of 3)**
>
> We appreciate the Reviewer XSmN for the detailed comments. Here we provide the clarifications below.
>
> **W1**.	The paper attributes improved sample efficiency to the combination of ensemble, reset, and multi-agent translation invariance. However, given that the reset mechanism is adopted from prior work [1] and the empirical impact of the multi-agent translation invariance appears marginal in the results, the primary contribution of the paper seems reduced to the mere application of a standard ensemble technique.
>
> We have provided a comprehensive comparison of EnSet and MARR in Appendix D, and key differences are listed in Table 3. Although the Shrink & Perturb is from MARR, we show that the ensemble-based Shrink & Perturb is more effective than the Shrink & Perturb alone as shown in the ablation study of Figure 5. More importantly, one of our major contributions is that we, for the first time, found that high replay ratios cause severe dormant neurons in the centralized global Q-network and the ensemble could effectively mitigate this with Shrink & Perturb. Another primary contribution of us is the multiagent translation invariance for diversifying experiences. Although this looks in a simple form, it is effective, and combining it with ensemble-based Shrink & Perturb boosts various MARL algorithms. Therefore, the findings of severe dormant neurons, the ensemble-based Shrink & Perturb to tackle it, as well as multiagent translation invariance to overcome overfitting at high replay ratios are all our main contributions.
>
> **W2**.	While achieving strong performance with fewer environment steps is intriguing, this approach, coupled with the ensemble method and a high replay ratio, likely incurs a substantial computational overhead (e.g., training time). A thorough analysis quantifying the computational cost vs. performance gain trade-off is currently missing and required for a complete evaluation.
>
> Here, we provide the computational cost of EnSet-QMIX with a replay ratio of 10 and the standard QMIX with a replay ratio of 1.
>
> | Scenario            | EnSet-QMIX (RR=10) | QMIX (RR=1) |
> |---------------------|--------------------|-------------|
> | 2s3z (0.5M steps)   |    5.84h   |   2.91h    |
> | 5m_vs_6m (1M steps) |     13.47h     |     6.13h     |
>
> At the same time, we report the CPU/GPU utilization in 2s3z. CPU utilization is around 100% of 1 core for both EnSet-QMIX and QMIX. Peak GPU utilization is 22% for EnSet-QMIX and 11% for QMIX. GPU memory usage is 546MiB for QMIX and 646MiB for EnSet-QMIX. We see that, although the replay ratio is improved from 1 to 10, the training time only increases roughly twice and other CPU/GPU utilization metrics also increase far less than 10 times.
>
> **W3**.	The overall clarity of the paper's narrative is inconsistent in specific sections: (1) The purpose and relevance of the content related to Equation 1 in Section 2.2 are ambiguous, particularly as the concept is not subsequently utilized, suggesting it may be superfluous. (2) The third paragraph of Section 4.1 requires significant revision for structural clarity and flow.
>
> a)	Equation 1 in Section 2.2 formally shows that to improve the sample efficiency, we can improve the replay ratio $N_{RR}$, which is our goal throughout this paper. $N_{RR}$ in Equation 1 is utilized subsequently. For $ \mathbb{E}[N_{sampled}]$, we use the words “sample efficiency” subsequently for a better understanding. We have revised the expression for Equation 1 to point out that its purpose is to “clarify the critical role of the replay ratio”.
>
> b)	Thanks to the reviewer for the suggestion. We have revised it accordingly to three paragraphs for better structural clarity and flow.
>
> **W4**.	The periodic 'Shrink & Perturb' operation is not clearly defined. It is uncertain whether this mechanism is applied to each individual network within the ensemble or to the ensemble as a whole, which hinders a complete understanding of the algorithm's functional mechanism.
>
> The periodic 'Shrink & Perturb' operation is applied to each individual network within the ensemble as described in Line 17 of Algorithm 1. Here we also understand “to the ensemble as a whole” the same as “to each individual network within the ensemble”. When 'Shrink & Perturb' the ensemble of global Q-networks, we 'Shrink & Perturb' every global Q-network in the ensemble.

---

> ### Author Response · Authors · 2025-11-24
> **Rebuttal part 2 (out of 3)**
>
> **W5**.	The experimental results lack justification for using different evaluation metrics (median with 25-75% percentiles and average with 95% confidence interval) across various experiments. Furthermore, the excessive smoothing applied to several plots significantly diminishes the graphical fidelity and reliability of the presented data.
>
> We follow the common settings for evaluation metrics. In SMAC, the evaluation metric is median performance with 25-75% percentiles [1]. In MPE, the evaluation metric is average with a 95% confidence interval [2]. The smooth weight is 0.8 (last * 0.8 + now * 0.2) for both SMAC and MPE, which we believe is a moderate value to balance the tendency of algorithms and keep the reliability of data.
>
> References
>
> [1] Samvelyan, Mikayel, et al. "The starcraft multi-agent challenge." arXiv preprint arXiv:1902.04043 (2019).
>
> [2] Peng, Bei, et al. "Facmac: Factored multi-agent centralised policy gradients." Advances in Neural Information Processing Systems 34 (2021): 12208-12221.
>
> **W6**.	A comprehensive table summarizing all crucial experimental setup parameters and algorithmic hyperparameters would greatly enhance the reproducibility and clarity of the work.
>
> All the special hyperparameters of EnSet are provided in Appendix B. Following the reviewer’s suggestion, we have now added a corresponding table (Table 1) in Appendix B. At the same time, other hyperparameters except the replay ratio follow the standard MARL algorithms.
>
> **Q1**.	Figures 2 and 4 suggest that merely reducing dormant neurons is insufficient to guarantee performance improvement. Could the authors elaborate on why this occurs, and, in light of these specific control results, what is the mechanism by which EnSet successfully translates reduced dormant neurons into superior performance?
>
> a)	First, in Figure 4, we have provided some analysis – “It is clear that EnSet maintains the dormant neuron rates at a low level and achieves the best performance among baselines. Interestingly, although MARR, ReDo, and ReBorn have lower dormant neuron rates than EnSet in MMM2 and 3s5z\_vs\_3s6z, they perform worse than EnSet. This indicates that, while a high dormant neuron rate prevents MARL from stabilizing learning, forcing a much lower dormant neuron rate may constrain network parameters and hurt network representation ability.”
>
> b)	Second, we believe keeping neuron information is important. Fully reset the dormant neuron may cause information loss. The Shrink & Perturb form in EnSet allows interpolation between current network parameters and an initialized parameter vector on each reset, which could be regarded as keeping part of the information as a soft reset. At the same time, the ensemble operation also naturally reduces the dormant neurons, while allowing information to be stored in different individual global Q-networks separately.
>
> **Q2**.	Considering the goal of mitigating dormant neurons, the 'redo' [2] mechanism has been explored in related literature as an alternative to 'reset'. Could the authors provide experimental results showing the performance of the proposed algorithm when the 'Shrink & Perturb' component is replaced solely with the 'Redo' technique while maintaining other components?
>
> Dear reviewer, in Figure 4, we have provided the comparison of EnSet with ReDo, and its updated version in MARL, ReBorn [1]. We found that both ReDo and ReBorn are not suitable for the high-replay-ratio setting in MARL.
>
> References
>
> [1] Qin, Haoyuan, et al. "The dormant neuron phenomenon in multi-agent reinforcement learning value factorization." Advances in Neural Information Processing Systems 37 (2024): 35727-35759.
>
> **Q3**.	Could the authors quantify the difference in computational cost (e.g., wall-clock time, GPU/CPU utilization) between the baseline algorithm and the proposed algorithm?
>
> Please see our response to weakness 2.

---

> ### Author Response · Authors · 2025-11-24
> **Rebuttal part 3 (out of 3)**
>
> **Q4**.	Previous research [3] in single-agent RL utilizes ensemble methods and reset mechanisms, demonstrating improved sample efficiency as the replay ratio increases, which feels similar to EnSet. What is the authors' opinion on comparing EnSet with this method?
>
> We appreciate the reviewer pointing out [3], and we have added this citation to our paper. At the same time, we list some key differences here for clarification.
>
> a)	First, [3] uses a hard reset to N-ensemble agents and resets each agent in the ensemble sequentially with adaptive composition to mitigate performance collapses. They do not connect the dormant neuron issue to the ensemble. On the contrary, the proposed EnSet for the first time connects the dormant neuron phenomenon with a high replay ratio in the centralized global Q-value network, while [3] does not study the dormant neuron. This motivation fundamentally differs our work from [3], which utilizes an ensemble to mitigate performance collapses. Therefore, part of our novelty relies on using an ensemble to reduce severe dormant neurons at a high replay ratio in MARL, and we are the first work to do this (in MARL).
>
> b)	Second, [3] studies single-agent safe RL while we focus on the general MARL. We specially consider the CTDE (centralized training with decentralized execution) setting for MARL, which raises a lot of unique questions different from the single-agent case. For example, multi-agent translation invariance is a kind of multiagent experience augmentation for the centralized Q-network.
>
> c)	To summarize, we believe the new motivation of the ensemble global Q-network to reduce dormant neurons at high replay ratio and the new multiagent experience augmentation technique (i.e., multiagent experience augmentation) are sufficient to prove the novelty of our paper, together with the strong empirical results.
>
> References
>
> [3] Kim et al., “Sample-Efficient and Safe Deep Reinforcement Learning via Reset Deep Ensemble Agents,” NeurIPS 2023

---

> > ### Comment · Reviewer_XSmN · 2025-11-27
> >
> > I appreciate the detailed response.
> > However, despite your comprehensive explanation, I still believe the paper faces the following issues.
> >
> > **W1**. Regarding the contribution, I still find it difficult to agree with the authors' claims.
> > Existing studies consistently indicate that a high replay ratio leads to plasticity loss or dormant neurons.
> > Considering that this is a fundamental issue stemming from the use of neural networks, regardless of whether the setting is single-agent or multi-agent, the distinctiveness of the proposed contribution regarding the claim that "*for the first time, found that high replay ratio causes severe dormant neurons in the centralized global Q-network*" remains unclear.
> >
> > **W2**. Given the use of an ensemble of 5 and a high replay ratio, it is surprising that the computational cost increases by only a factor of two compared to QMIX. In my experience, computational cost typically scales linearly with the number of ensembles and the replay ratio. I question why this scaling logic does not apply in this case.
> >
> > **W3**. Even after your explanation, I remain unconvinced of the necessity of this metric in the paper.
> > According to your explanation and Equation 1, a high replay ratio ultimately increases sample efficiency.
> > However, doesn't this contradict the authors' core premise that a high replay ratio induces dormant neurons, thereby reducing sample efficiency?
> > I fail to see the justification for retaining this section, as it may invite such confusion.
> >
> > **Q2**. To clarify, what I requested was the result of replacing only the 'shrink & perturb' mechanism with 'ReDo', while retaining all other components of EnSet.
> > I am interested in the performance difference when only the 'shrink & perturb' is swapped.
> > In my view, if the primary issue is dormant neurons, as the authors claim, the model should be able to achieve comparable performance by simply utilizing 'ReDo' under the same conditions.

---

> > > ### Author Response · Authors · 2025-12-01
> > > **Further rebuttal for Reviewer XSmN part 1 (out of 2)**
> > >
> > > We are very glad that we have addressed most of the Reviewer XSmN’s concerns and questions. We appreciate the Reviewer XSmN for the further detailed comments. Here we provide the clarifications below.
> > >
> > > **W1**. Regarding the contribution, I still find it difficult to agree with the authors' claims. Existing studies consistently indicate that a high replay ratio leads to plasticity loss or dormant neurons. Considering that this is a fundamental issue stemming from the use of neural networks, regardless of whether the setting is single-agent or multi-agent, the distinctiveness of the proposed contribution regarding the claim that "for the first time, found that high replay ratio causes severe dormant neurons in the centralized global Q-network" remains unclear.
> > >
> > > Before our work, no work found that a high replay ratio leads to severe dormant neurons in the domain of MARL. The scientific exploration needs to be explored step-by-step. Even if some works in the single-agent domain found that a high replay ratio leads to plasticity loss or dormant neurons, the situation in MARL remains unclear as the multiagent case has its own features and challenges. For example, as shown in Figure 9 of Appendix A, when the replay ratio is high, the dormant neurons in the agent network decrease. This contradicts the general findings in the single-agent case, and necessitates that we should handle the severe dormant neurons in the global Q-network when the replay ratio is high. This phenomenon further confirms our contribution that “for the first time, found that high replay ratio causes severe dormant neurons in the centralized global Q-network”.
> > >
> > > **W2**. Given the use of an ensemble of 5 and a high replay ratio, it is surprising that the computational cost increases by only a factor of two compared to QMIX. In my experience, computational cost typically scales linearly with the number of ensembles and the replay ratio. I question why this scaling logic does not apply in this case.
> > >
> > > The running time is averaged over 6 seeds, and each run is with enough computational resources (CPU, GPU, and memory). Then let us explain why scaling logic does not apply.
> > >
> > > First, for the ensemble size, we implement it with the batch mode in PyTorch instead of a time-consuming “for” loop. Specifically, the global Q-value is computed as a vector of [batch size, episode length, ensemble size]. Therefore, even when the ensemble is larger than 1, its computation only needs a single forward and only increases a small portion of time.
> > >
> > > Second, for the high replay ratio, we need to notice that the total running time of an RL algorithm could be roughly divided into three parts: the environment interaction time, the training time, and other time for the rest code. Increasing the replay ratio will not influence the environment interaction time (such as inferring actions from policies) and other time for rest codes (such as logging). Increasing the replay ratio will only scale up the training time linearly. In the multiagent SMAC case, the environment interaction needs some time to execute actions and return states as its simulation is complex. Therefore, we see that, although the replay ratio is improved from 1 to 10, the training time only increases roughly twice. So, the complexity of the environment needs to be taken into consideration.
> > >
> > > Finally, we want to stress that our main goal/contribution is to improve the sample efficiency to achieve the same (or better) performance with fewer environment interactions. This is vital in realistic scenarios where the environment interaction is limited or the simulation is expensive with sophisticated system dynamics. In these cases, the training time with CPU/GPU (which will also become cheaper and cheaper in cost) is of less importance.

---

> > > > ### Author Response · Authors · 2025-12-01
> > > > **Further rebuttal for Reviewer XSmN part 2 (out of 2)**
> > > >
> > > > **W3**. Even after your explanation, I remain unconvinced of the necessity of this metric in the paper. According to your explanation and Equation 1, a high replay ratio ultimately increases sample efficiency. However, doesn't this contradict the authors' core premise that a high replay ratio induces dormant neurons, thereby reducing sample efficiency? I fail to see the justification for retaining this section, as it may invite such confusion.
> > > >
> > > > Dear reviewer, there exists an implicit condition that after improving the replay ratios, the performance should also improve. Simply increasing the replay ratio does not increase the sample efficiency, as it induces severe dormant neurons. Therefore, we need new techniques such as EnSet to tackle the severe dormant neurons at high replay ratios to improve performance, and finally improve sample efficiency. As EnSet achieves a better performance than the standard MARL algorithms with the same environment steps, we say that EnSet achieves a better sample efficiency by training MARL at high replay ratios. To make our expression clear, we now add the following sentence after the Equation 1 - “Typically, works focusing on training reinforcement learning agents at a high replay ratio \citep{chen_randomized_2021} aim to increase $N_{RR}$ while keeping $N_{B}$, $V$, and $T_{U}$ unchanged. They seek to achieve higher sample efficiency through a higher $\mathbb{E}[N_{sampled}]$ with different approaches, finally improving performance with the same number of environment interactions, i.e., the same number of collected data experiences.” (The original expression – “Typically, works focusing on training reinforcement learning agents at a high replay ratio \citep{chen_randomized_2021} aim to improve $N_{RR}$ while keeping $N_{B}$, $V$, and $T_{U}$ unchanged to reach a higher sample efficiency indicated by $\mathbb{E}[N_{sampled}]$.” We are aware that this original expression has an implicit condition that after improving the replay ratios, the performance also improves. We have revised it to the new expression.)
> > > >
> > > > **Q2**. To clarify, what I requested was the result of replacing only the 'shrink & perturb' mechanism with 'ReDo', while retaining all other components of EnSet. I am interested in the performance difference when only the 'shrink & perturb' is swapped. In my view, if the primary issue is dormant neurons, as the authors claim, the model should be able to achieve comparable performance by simply utilizing 'ReDo' under the same conditions.
> > > >
> > > > Dear reviewer, thanks for the clarification. We have now added the experiments by replacing only the 'shrink & perturb' mechanism in EnSet with 'ReDo', ‘ReBorn’, and ‘ReSet’ in Figure 15 in Appendix I. We say that, when the EnSet-version ‘ReDo’ combines with the global Q-network ensemble and global Q-function translation invariance, it also achieves a good performance at a high replay ratio of 10. However, the EnSet-version ‘ReDo’ still performs worse than EnSet (with 'Shrink & Perturb').
> > > >
> > > > At the same time, we see that all the EnSet-based reset techniques reduce the severe dormant neurons compared with the original ‘Vanilla’ QMIX. Although the EnSet-version ‘ReDo’ (also EnSet-‘ReSet’ and EnSet-‘ReBorn’) has a lower dormant neuron rate than EnSet (with 'Shrink & Perturb'), it performs worse. This indicates that forcing a much lower dormant neuron rate may constrain network parameters and hurt network representation ability, which is also observed in Figure 4 in Section 4.2.
> > > >
> > > > These results show that the 'Shrink & Perturb' is compatible with EnSet best among these reset techniques, which may benefit from its soft-reset mechanism.

---

### Official Review · Reviewer_dB9H · 2025-10-24

**Soundness:** 3
**Presentation:** 3
**Contribution:** 3
**Rating:** 6
**Confidence:** 3

**Summary:**

The paper addresses the challenge of improving sample efficiency in multiagent reinforcement learning (MARL) by utilizing high replay ratios. It introduces a novel method called Ensemble Reset (EnSet), which combines an ensemble of global Q-value networks with parameter resets to mitigate the issue of dormant neurons that arise when training at high replay ratios. The authors assert that this approach enhances the training efficiency of various MARL algorithms in environments such as SMAC (StarCraft Multi-Agent Challenge) and MPE (Multiagent Particle Environment).

**Strengths:**

1. This paper provides a new perspective on enhancing sample efficiency by addressing the dormant neuron phenomenon
2. The authors conduct extensive experiments across multiple environments (SMAC, MPE, and SMACv2) and a variety of tasks, demonstrating the effectiveness of EnSet. The results show that EnSet significantly improves the performance of existing MARL algorithms at high replay ratios.
3. The identification and analysis of the dormant neuron phenomenon in the global Q-network is a crucial finding. The paper effectively links this issue to the challenges of high replay ratios, providing a clear motivation for the proposed solution.

**Weaknesses:**

1. The distinction between MARR and EnSet is not clearly explained. It seems like an ensemble version of MARR with added translation, but the ablation study shows that translation does not contribute significantly.
2. It’s not clear that whether the dormant neuron phenomenon only occur in the global Q-network.

**Questions:**

1. Please provide a more detailed explanation of the key distinctions between MARR and EnSet, particularly regarding their motivations, design choices, and the specific problems they aim to solve.
2. We recommend adding ATM(RR=10) and similar baselines to Figure 3 for direct comparison.
3. Why do all the comparison methods in Figure 4 choose to build on QMIX rather than ATM? We recommend adding this experiments.

---

> ### Author Response · Authors · 2025-11-25
> **Rebuttal**
>
> We appreciate the Reviewer dB9H for the detailed comments. Here we provide the clarifications below.
>
> **W1** The distinction between MARR and EnSet is not clearly explained. It seems like an ensemble version of MARR with added translation, but the ablation study shows that translation does not contribute significantly.
>
> We have provided the distinction between MARR and EnSet in the Appendix by directly listing the key differences in Table 3 and conducting more comparisons of their standard versions in Figure 10. For the technique part, EnSet could be regarded as an updated version of MARR with a global Q-network ensemble and global Q-function translation invariance. More importantly, for the motivation part, EnSet for the first time found that the severe dormant neurons in the global Q-network exist at high replay ratios, which hinders the sample efficiency of MARL. Therefore, one major contribution of EnSet is to explicitly investigate the connection between dormant neurons in the global Q-network and high replay ratio.
>
> **W2** It’s not clear that whether the dormant neuron phenomenon only occur in the global Q-network.
>
> Dear reviewer, thank you for pointing out this! Now we have added the dormant neuron in the agent network in Figure 9 of Appendix A with more descriptions. We see that the dormant neuron rates of the agent network maintain a low level even when the replay ratio is high. Therefore, the severe dormant neurons happen in the global Q-network at high replay ratios, while the agent networks do not suffer from this.
>
> **Q1** Please provide a more detailed explanation of the key distinctions between MARR and EnSet, particularly regarding their motivations, design choices, and the specific problems they aim to solve.
>
> Please refer to our response in W1 for key distinctions between MARR and EnSet. And we elaborate with more explanations below.
>
> For motivations, MARR aims to address the primacy bias to train MARL at high replay ratios. However, they do not explicitly study the deeper reason for primacy bias in MARL. Differently, EnSet aims to address the dormant neurons in the global Q-network to train MARL at high replay ratios, which fundamentally reveals that severe dormant neurons exist in the global Q-network at high replay ratios.
>
> For design choice, EnSet could be regarded as an updated version of MARR with a global Q-network ensemble and global Q-function translation invariance. Moreover, EnSet uses a series environment, as most standard MARL algorithms while MARR uses parallel environments.
>
> For the specific problem, both algorithms aim to train MARL algorithms at high replay ratios to improve sample efficiency.
>
> **Q2** We recommend adding ATM(RR=10) and similar baselines to Figure 3 for direct comparison.
>
> Experiments in Figure 3 show that EnSet is able to boost the performance of popular MARL algorithms (the standard ATM, QPLEX, and QMIX with a replay ratio of 1) with high replay ratios. For the standard baselines, increasing the replay ratio from 1 to a high value such as 10 will destroy the learning of them, as shown in Figure 2(a)-(d) and Figure 12 in Appendix F. For example, in Figure 12, we see RR=10 is better for EnSet-QMIX while RR=1 is better for the standard QMIX, so we believe this is a fair comparison by showing the standard MARL (e.g., ATM) with RR = 1 and EnSet-based MARL (e.g., EnSet-ATM) with RR = 10 in Figure 3.
>
> **Q3** Why do all the comparison methods in Figure 4 choose to build on QMIX rather than ATM? We recommend adding this experiments.
>
> Experiments in Figure 4 compare EnSet with other MARL network reset methods to show that EnSet is the superior reset technique for MARL at high replay ratios. The reason why we use QMIX instead of ATM as the base algorithm is that QMIX is one of the most popular and widely-used MARL algorithms for the baseline and backbone algorithm in the MARL domain (Previous reset techniques for MARL also use QMIX as the base algorithm such as [1] and [2]). At the same time, our goal is to compare different MARL reset techniques. We use the standard setting of QMIX as the base algorithm. If we use the ATM as the base algorithm, more reviewers will ask why not use QMIX as the base algorithm.
>
> References
>
> [1] Qin, Haoyuan, et al. "The dormant neuron phenomenon in multi-agent reinforcement learning value factorization." Advances in Neural Information Processing Systems 37 (2024): 35727-35759.
>
> [2] Yang, Yaodong, Guangyong Chen, and Pheng-Ann Heng. "Sample-efficient multiagent reinforcement learning with reset replay." Forty-first international conference on machine learning. 2024.

---

> > ### Comment · Reviewer_dB9H · 2025-11-25
> >
> > Thank you for the reply and clarifications. I would like to maintain my score.

---

> > > ### Author Response · Authors · 2025-12-01
> > > **Thanks for the Reviewer dB9H**
> > >
> > > Dear Reviewer dB9H, thank you so much for your positive evaluation and feedback! Your comments and suggestions are greatly valuable to our work!

---

### Official Review · Reviewer_33vY · 2025-10-27

**Soundness:** 3
**Presentation:** 3
**Contribution:** 2
**Rating:** 4
**Confidence:** 4

**Summary:**

This paper investigates training MARL algorithms under high replay ratios. The authors identify that increasing the replay ratio exacerbates the dormant neuron phenomenon in the centralized global Q-network, which leads to representational collapse and unstable learning. To address this, they propose EnSet, combining an ensemble of global Q-networks with periodic parameter reset (Shrink & Perturb) and a data augmentation method based on multi-agent translation invariance.
Comprehensive experiments across SMAC, MPE, and SMACv2 demonstrate that EnSet stabilizes training and improves sample efficiency for various MARL algorithms at high replay ratios.

**Strengths:**

1. The paper addresses an important problem: how to effectively scale MARL to higher replay ratios for improved sample efficiency, which is increasingly relevant for large-scale and expensive environments.

2. Experiments span multiple benchmarks and show consistent improvements across algorithms and replay ratios. The dormant neuron visualizations and gradient analyses are insightful and help justify the design choices.

3. The paper connects dormant neuron growth with high replay ratios and empirically shows that ensembles and resets mitigate this effect.

**Weaknesses:**

1. Overstated novelty

The dormant neuron issue in QMIX has already been discussed in [1]. Although this paper focuses on high replay ratios, it is not accurate to claim being the first to identify dormant neurons in global Q-networks.
The Shrink & Perturb reset mechanism originates from [2]. This paper reuses the idea without substantial conceptual innovation.
In addition [3] also demonstrates that ensembles can improve QMIX’s performance in SMAC and SMACv2.

2. Limited training horizon and convergence analysis

Most experiments stop at 1M environment steps to emphasize sample efficiency, but many methods (including the baselines) have not yet converged. Showing converged performance or stability at longer horizons would help assess whether EnSet maintains strong final performance and robustness rather than only accelerating early learning.

3. Framework inconsistency

The paper adopts pymarl2 for SMACv2 experiments but uses the older pymarl implementation for SMAC. Since pymarl2 provides stronger and more optimized baselines, the comparison on SMAC may not fairly reflect the advantage of the proposed method.

4. Unclear analysis of ensemble benefits

The paper states that ensembles are applied only to global Q-networks because local Q-networks do not suffer from dormant neurons. However, in experiments, MADDPG’s critic (which corresponds to individual Q-value) also benefits from ensembles.
This raises an important open question: is the dormant neuron phenomenon primarily due to high-dimensional joint action/state input or replay ratio scaling? A deeper analysis of when and where the problem arises would strengthen the contribution.

[1] The Dormant Neuron Phenomenon in Multi-Agent Reinforcement Learning Value Factorization

[2] Sample-Efficient Multiagent Reinforcement Learning with Reset Replay

[3] Revisiting Cooperative Off-Policy Multi-Agent Reinforcement Learning

**Questions:**

1. How significant is the computational overhead of EnSet (e.g., ensemble size H=5) compared to standard QMIX or MARR?

2. In the main experiment results, does MARR use the same parallel environments with EnSet?

---

> ### Author Response · Authors · 2025-11-25
> **Rebuttal part 1 (out of 2)**
>
> We appreciate the Reviewer 33vY for the detailed comments. Here we provide the clarifications below.
>
> **W1.	(Overstated novelty)** The dormant neuron issue in QMIX has already been discussed in [1]. Although this paper focuses on high replay ratios, it is not accurate to claim being the first to identify dormant neurons in global Q-networks. The Shrink & Perturb reset mechanism originates from [2]. This paper reuses the idea without substantial conceptual innovation. In addition [3] also demonstrates that ensembles can improve QMIX’s performance in SMAC and SMACv2.
>
> a)	First, [1] and [3] do not study the high-replay-ratio setting. We do not claim “being the first to identify dormant neurons in global Q-networks”. We cite our original expression – “First and for the first time, we reveal that the centralized global Q-network of MARL suffers from a severe dormant neuron issue at high replay ratios, where a large portion of network neurons become inactive to undermine network expressivity.” Here, we only claim that we are the first to find that the centralized global Q-network suffers from a severe dormant neuron issue at high replay ratios.
>
> b)	Second, [2] and [3] do not study the dormant neuron problem. At the same time, we have listed key comparisons with [2] in Appendix D. Comparison with [2], we propose the ensemble to tackle severe dormant neurons at high replay ratios and multiagent translation invariance to diversify experience, finally to improve sample efficiency. Ensemble in MARL ([3] for reducing variance, [4] for reducing overestimation, [5] for exploration) has been studied for different purposes. Comparing with [3], here we introduce an ensemble of global Q-networks to tackle severe dormant neurons at high replay ratios with a different purpose.
>
> References
>
> [1] The Dormant Neuron Phenomenon in Multi-Agent Reinforcement Learning Value Factorization
>
> [2] Sample-Efficient Multiagent Reinforcement Learning with Reset Replay
>
> [3] Revisiting Cooperative Off-Policy Multi-Agent Reinforcement Learning
>
> [4] Yang, Yaodong, et al. "Dual Ensembled Multiagent Q-Learning with Hypernet Regularizer." AAMAS 2025.
>
> [5] Lukas Schäfer, et al. “Ensemble Value Functions for Efficient Exploration in Multi-Agent Reinforcement Learning.” AAMAS 2025.
>
> **W2.	(Limited training horizon and convergence analysis)** Most experiments stop at 1M environment steps to emphasize sample efficiency, but many methods (including the baselines) have not yet converged. Showing converged performance or stability at longer horizons would help assess whether EnSet maintains strong final performance and robustness rather than only accelerating early learning.
>
> In Appendix H and Figure 14, we have provided a comparison of EnSet with standard MARL with more environment steps. It shows that, even though the number of environment interaction steps of the standard MARL is twice that of EnSet in 3s5z vs 3s6z and corridor and four times that of EnSet in pp9a, EnSet’s performance is still better than the standard algorithms in terms of the final policy performance, showing EnSet's impressive sample efficiency.
>
> **W3.	(Framework inconsistency)** The paper adopts pymarl2 for SMACv2 experiments but uses the older pymarl implementation for SMAC. Since pymarl2 provides stronger and more optimized baselines, the comparison on SMAC may not fairly reflect the advantage of the proposed method.
>
> In both pymarl2 for SMACv2 and pymarl for SMAC, we keep the training and evaluation settings the same across methods. As we do not utilize any additional tricks from pymarl2 for EnSet in SMAC when compared with baselines, we believe the comparison is fair. At the same time, when EnSet-QMIX and QMIX are with optimized tricks in pymarl2 for SMACv2, EnSet also boosts the optimized QMIX, demonstrating the effectiveness of our method.

---

> > ### Author Response · Authors · 2025-11-25
> > **Rebuttal part 2 (out of 2)**
> >
> > **W4.	(Unclear analysis of ensemble benefits)** The paper states that ensembles are applied only to global Q-networks because local Q-networks do not suffer from dormant neurons. However, in experiments, MADDPG’s critic (which corresponds to individual Q-value) also benefits from ensembles. This raises an important open question: is the dormant neuron phenomenon primarily due to high-dimensional joint action/state input or replay ratio scaling? A deeper analysis of when and where the problem arises would strengthen the contribution.
> >
> > Here, we focus on the cooperative MARL setting (all agents share the same reward) and consider the centralized training with decentralized execution (CTDE) paradigm. Therefore, here the MADDPG’s critic is a centralized critic (also could be called as the global Q-network to predict the global Q-value as $Q_{tot}(s, a_{1}, …, a_{N})$). This centralized critic takes the state input consists of the global state and actions of agents without view restriction. For more details of implementation, please refer to the paper and codebase of [1] which we follow in MPE experiments.
> >
> > At the same time, we agree with you that a deeper analysis of when and where the severe dormant neuron problem arises is important. Currently, our paper shows that the severe dormant neuron problem arises in the global Q-network at high replay ratios and proposes a solution in this situation to boost MARL. We have added it to the further work for a further deep study of dormant neurons in MARL – “Third, a deeper and theoretical analysis of how the dormant neuron phenomenon in MARL arises and proceeds during training is also important.”.
> >
> > References
> >
> > [1] Peng, Bei, et al. "Facmac: Factored multi-agent centralised policy gradients." Advances in Neural Information Processing Systems 34 (2021): 12208-12221.
> >
> > **[Q1]** How significant is the computational overhead of EnSet (e.g., ensemble size H=5) compared to standard QMIX or MARR?
> >
> > Here we provide the computational cost of EnSet-QMIX (ensemble size H=5) with a replay ratio of 10, the standard QMIX with a replay ratio of 1, and MARR with a replay ratio of 10.
> >
> > | Scenario | EnSet-QMIX (RR=10) | QMIX (RR=1) | MARR-QMIX (RR=10) |
> > |---------------------|--------------------|-------------|-------------|
> > | 2s3z (0.5M steps)   |    5.84h   |   2.91h    |  5.50h |
> > | 5m_vs_6m (1M steps) |  13.47h  | 6.13h     | 12.42h |
> >
> > We see that, although the replay ratio is improved from 1 to 10, the training time is only roughly twice for EnSet.
> >
> > **[Q2]** In the main experiment results, does MARR use the same parallel environments with EnSet?
> >
> > In the main experiment results, all the methods, including EnSet and MARR, use the same series environment setting instead of the parallel environments. At the same time, we also notice that in MARR’s original paper, the authors use parallel environments. Therefore, in Appendix D, we compare EnSet with the original MARR, which enables the parallel environments. Figure 10 in Appendix D shows that the standard EnSet (series environment, replay ratio of 10) outperforms the standard MARR (8 parallel environments, replay ratio of 50).

---

### Official Review · Reviewer_UboR · 2025-10-27

**Soundness:** 1
**Presentation:** 2
**Contribution:** 1
**Rating:** 2
**Confidence:** 4

**Summary:**

This paper proposes a method that uses an ensemble of global Q-networks to increase the replay ratio while avoiding the problem of dormant neurons. In addition, the authors adopt an augmentation technique called multi-agent translation invariance to diversify the replay experience.

**Strengths:**

This paper tackles an important aspect of deep MARL regarding sample efficiency, and I believe this line of research is important.

**Weaknesses:**

**(Lack of novelty)**

The contributions include having multiple hypernetworks (called the global Q-network) and introducing multi-agent translation invariance. I understand from the ICLR review guideline that creative combinations of existing ideas can be a good contribution. However, naively using an ensemble network is not novel at all. In addition, there is a work in the single-agent RL domain [1], which further reduces the novelty of this work. Even though [1] is in the single-agent setting, [1] and this work are highly related in the sense that both use an ensemble of networks to allow a high replay ratio. Moreover, this work uses an ensemble of global Q-networks—which can actually be seen as a single agent (since centralized methods treat multiple agents as a single agent). The authors should mention [1] in the related work section.

**(Lack of clarification of multi-agent translation invariance)**

 It is hard to understand the motivation and effectiveness of translation invariance. It looks to me like injecting noise (z is sampled uniformly). Why does it diversify the replay experience? This is a common way to improve robustness. How is it related to improving sample efficiency and preventing dormant neurons? The authors should provide a clear motivation and results on this.

**(Inefficient and unclear experiments)**

– The experiments are generally not well organized. For example, Figure 3 shows the performance with and without EnSet on top of ATM, QPLEX, and QMIX, and it seems ATM outperforms the others in all tasks. It is good to have the results of many baselines. However, Figure 4 only shows the results of all reset methods and the proposed method on top of QMIX. I believe this is the primary result showing the contribution, but why do the authors not use ATM as the base algorithm? Basically, the algorithm performing best is EnSet-ATM, and the primary baseline is MARR, so the result of MARR-ATM must be included. In addition, Figure 3 shows the result of ATM when RR = 1 and EnSet-ATM when RR = 10, which is unfair (I acknowledge that EnSet-ATM when RR = 1 is provided in the appendix, but the authors should be careful about what results are included in the main paper).

– Again, I appreciate that the authors use different base algorithms. However, in my opinion, the authors use different baselines in terms of reset methods, not RL algorithms—for example, EnSet-Reborn vs. Reborn or ReDo-Reborn vs. ReDo. This also applies to MPE and SMACv2. The results that EnSet-FACMAC is better than FACMAC in MPE and EnSet-QMIX (RR = 15) is better than QMIX (RR = 1) are not enough. The results of other reset methods must be included. Again, since the contribution is using an ensemble method, the primary baselines should be other reset methods, not just naïve MARL algorithms.

[1] Kim et al., “Sample-Efficient and Safe Deep Reinforcement Learning via Reset Deep Ensemble Agents,” NeurIPS 2023

**Questions:**

See Weaknesses

---

> ### Author Response · Authors · 2025-11-21
> **Rebuttal part 1 (out of 3)**
>
> We appreciate the Reviewer UboR for the detailed comments. Here we provide the clarifications below.
>
> **1.	(Lack of novelty)** The contributions include having multiple hypernetworks (called the global Q-network) and introducing multi-agent translation invariance. I understand from the ICLR review guideline that creative combinations of existing ideas can be a good contribution. However, naively using an ensemble network is not novel at all. In addition, there is a work in the single-agent RL domain [1], which further reduces the novelty of this work. Even though [1] is in the single-agent setting, [1] and this work are highly related in the sense that both use an ensemble of networks to allow a high replay ratio. Moreover, this work uses an ensemble of global Q-networks—which can actually be seen as a single agent (since centralized methods treat multiple agents as a single agent). The authors should mention [1] in the related work section.
>
> a)	First, the proposed EnSet for the first time connects the dormant neuron with a high replay ratio in the centralized global Q-value network, while [1] does not study the dormant neuron phenomenon. This motivation fundamentally differs our work from [1], which utilizes an ensemble to mitigate performance collapses. On the contrary, part of our novelty relies on using an ensemble of the global Q-network to reduce the severe dormant neurons at a high replay ratio in MARL. Although this seems simple, it is quite efficient, and we are the first to do this (in MARL).
>
> b)	Second, we must say that CTDE (centralized training with decentralized execution) is not a simple single-agent case. The centralized multiagent Q-network raises a lot of unique questions different from the single-agent case such as credit assignment, network design, and even experience augmentation from the perspective of multiagent systems. Specifically, multi-agent translation invariance is a kind of multiagent experience augmentation for the centralized Q-network by considering the multiagent system's property.
>
> c)	To summarize, the novelty does not rely on the complexity of the proposed technique. We believe the new motivation of the ensemble global Q-network to reduce dormant neurons at a high replay ratio and the new multiagent experience augmentation technique (i.e., multiagent experience augmentation) are sufficient to prove the novelty of our paper, together with the strong empirical results.
>
> d)	We appreciate the reviewer pointing out [1], and we have added this citation to our paper. At the same time, we list some key differences here for clarification.
>
> i.	Most importantly, [1] studies single-agent safe RL while we focus on the general MARL.
>
> ii.	[1] uses a hard reset to N-ensemble agents and resets each agent in the ensemble sequentially with adaptive composition to mitigate performance collapses caused by a hard reset. However, they never connect the dormant neuron issue to the ensemble. Our contribution first relies on finding the severe dormant neuron issue in the centralized global Q-network, and then we find that utilizing the global Q-ensemble is an effective way to address it.
>
> References
>
> [1] Kim et al., “Sample-Efficient and Safe Deep Reinforcement Learning via Reset Deep Ensemble Agents,” NeurIPS 2023

---

> ### Author Response · Authors · 2025-11-21
> **Rebuttal part 2 (out of 3)**
>
> **2.	(Lack of clarification of multi-agent translation invariance)** It is hard to understand the motivation and effectiveness of translation invariance. It looks to me like injecting noise (z is sampled uniformly). Why does it diversify the replay experience? This is a common way to improve robustness. How is it related to improving sample efficiency and preventing dormant neurons? The authors should provide a clear motivation and results on this.
>
> a)	As explained in Section 3.3, multiagent translation invariance is used to diversify the replay experience. Injecting noise would cause additional inaccuracy to the global Q-function. For example, if the agent health feature is added by +0.1, the global Q-value on this state will be positively biased from the true value and may provide wrong update signals. By equation (7) showing the global Q-function translation invariance, we diversify the experience without introducing additional inaccuracy.
>
> b)	As shown in [1] and the works of primacy bias, learning at a high replay ratio would make the agents incur a risk of overfitting to earlier experiences, negatively affecting the rest of the learning process. Therefore, we introduce the multiagent translation invariance to diversify experiences to improve sample efficiency, and finally to boost MARL at high replay ratios.
>
> References
>
> [1] Kim et al., “Sample-Efficient and Safe Deep Reinforcement Learning via Reset Deep Ensemble Agents,” NeurIPS 2023
>
> **3.	(Inefficient and unclear experiments)** – The experiments are generally not well organized. For example, Figure 3 shows the performance with and without EnSet on top of ATM, QPLEX, and QMIX, and it seems ATM outperforms the others in all tasks. It is good to have the results of many baselines. However, Figure 4 only shows the results of all reset methods and the proposed method on top of QMIX. I believe this is the primary result showing the contribution, but why do the authors not use ATM as the base algorithm? Basically, the algorithm performing best is EnSet-ATM, and the primary baseline is MARR, so the result of MARR-ATM must be included. In addition, Figure 3 shows the result of ATM when RR = 1 and EnSet-ATM when RR = 10, which is unfair (I acknowledge that EnSet-ATM when RR = 1 is provided in the appendix, but the authors should be careful about what results are included in the main paper).
>
> a)	(why do the authors not use ATM as the base algorithm?) Experiments in Figure 3 and experiments in Figure 4 have different purposes. Experiments in Figure 3 show that EnSet is able to boost the performance of popular MARL algorithms (ATM, QPLEX, and QMIX) under high replay ratios. Experiments in Figure 4 compare EnSet with other MARL network reset methods to show EnSet is the superior reset technique for MARL at high replay ratios. The reason why we use QMIX instead of ATM as the base algorithm is that QMIX is one of the most popular and widely-used MARL algorithms for a baseline and backbone algorithm in the MARL domain (Previous reset techniques for MARL also use QMIX as the base algorithm such as [2] and [3]). At the same time, our goal is to compare different MARL reset techniques. We use the standard setting of QMIX as the base algorithm. If we use the ATM as the base algorithm, more reviewers will ask why not use QMIX as the base algorithm.
>
> b)	(the result of MARR-ATM must be included) Figure 10 (a) and (b) in Appendix D provides the comparison of the EnSet-ATM and MARR-ATM. At the same time, Figure 10 (c) and (d) in Appendix D provide the comparison of the EnSet-FACMAC and MARR-FACMAC.
>
> c)	(ATM when RR = 1 and EnSet-ATM when RR = 10) Our main contribution is that EnSet boosts MARL algorithms at high replay ratios, compared with the standard versions of MARL algorithms. Therefore, we show ATM with RR = 1 and EnSet-ATM with RR = 10 in Figure 3. At the same time, as shown in Figure 12 in Appendix F, we see RR=10 is better for EnSet while RR=1 is better for the standard algorithm, so we believe this is a fair comparison by showing ATM with RR = 1 and EnSet-ATM with RR = 10 in Figure 3.
>
> References
>
> [2] Qin, Haoyuan, et al. "The dormant neuron phenomenon in multi-agent reinforcement learning value factorization." Advances in Neural Information Processing Systems 37 (2024): 35727-35759.
>
> [3] Yang, Yaodong, Guangyong Chen, and Pheng-Ann Heng. "Sample-efficient multiagent reinforcement learning with reset replay." Forty-first international conference on machine learning. 2024.

---

> ### Author Response · Authors · 2025-11-21
> **Rebuttal part 3 (out of 3)**
>
> **4.	(Inefficient and unclear experiments)** – Again, I appreciate that the authors use different base algorithms. However, in my opinion, the authors use different baselines in terms of reset methods, not RL algorithms—for example, EnSet-Reborn vs. Reborn or ReDo-Reborn vs. ReDo. This also applies to MPE and SMACv2. The results that EnSet-FACMAC is better than FACMAC in MPE and EnSet-QMIX (RR = 15) are better than QMIX (RR = 1) are not enough. The results of other reset methods must be included. Again, since the contribution is using an ensemble method, the primary baselines should be other reset methods, not just naïve MARL algorithms.
>
> a)	In Figure 4, we have benchmarked different MARL reset methods in SMAC. EnSet significantly outperforms other baselines as most of the MARL reset methods are not specifically designed for the high-replay-ratio setting. For MARR, which focuses on the high-replay-ratio setting, we compare it in Figure 10 in Appendix D. Our contribution is not only “using an ensemble method”. Our contribution is to propose the global Q-network ensemble reset to tackle severe dormant neurons and multiagent translation invariance in the global Q-function to diversify multiagent states, to ultimately boost various MARL algorithms at high replay ratios for sample efficiency. Therefore, we mainly demonstrate the performance of EnSet-based MARL and the standard MARL algorithms.

---

### Author Response · Authors · 2025-11-25
**New version of our paper during rebuttal**

Dear reviewers,

We have updated a new version of our paper during the rebuttal based on your comments and suggestions. You can read our rebuttal with the latest version, as we cite the contents in both the main paper and the Appendix for a lot of replies to your questions/concerns.

Best,

The authors

---

### Comment · Area_Chair_vSod · 2025-11-25

Dear Reviewers

Thank you for your time and help for reviews.
The author-reviewer discussion due is in one week. If you have not done yet, please review the authors' rebuttal for the paper under your evaluation and engage in discussion with authors.

Thank you again.
Best,

Area Chair

---

### Author Response · Authors · 2025-12-01
**A gentle reminder from the authors**

Dear AC and reviewers,

We have carefully answered each question and concern of each reviewer with detailed explanations and additional experiments.

Please check the latest version if you are convenient, as we cite the contents in both the main paper and the Appendix for a lot of replies to your questions/concerns.

Best,

The authors

---

### Note · Authors · 2026-01-08

I have read and agree with the venue's withdrawal policy on behalf of myself and my co-authors.